# MEXA: MULTILINGUAL EVALUATION OF ENGLISH-CENTRIC LLMS VIA CROSS-LINGUAL ALIGNMENT

## ABSTRACT

English-centric large language models (LLMs) often show strong multilingual capabilities. However, the multilingual performance of these models remains unclear and is not thoroughly evaluated for many languages. Most benchmarks for multilinguality focus on classic NLP tasks, or cover a minimal number of languages. We introduce MEXA, a method for assessing the multilingual capabilities of pre-trained English-centric LLMs using parallel sentences, which are available for more languages than existing downstream tasks. MEXA leverages the fact that English-centric LLMs use English as a kind of pivot language in their intermediate layers. It computes the alignment between English and non-English languages using parallel sentences to evaluate the transfer of language understanding from English to other languages. This alignment can be used to estimate task performance in other languages. We conduct studies using various parallel datasets (FLORES-200 and Bible), models (Llama family, Gemma family, Mistral, and OLMo), and established downstream tasks (Belebele, m-MMLU, and m-ARC). We explore different methods to compute embeddings in decoder-only models. Our results show that MEXA, in its default settings, achieves a statistically significant average Pearson correlation of 0.90 with three established downstream tasks across nine models and two parallel datasets. This suggests that MEXA is a reliable method for estimating the multilingual capabilities of English-centric LLMs, providing a clearer understanding of their multilingual potential and the inner workings of LLMs.

🤗 **Leaderboard** [anonymized url] 💻 **Code** [anonymized url]

## 1 INTRODUCTION

Most state-of-the-art autoregressive large language models (LLMs) are English-centric, closed-source models such as Claude 3 Opus, GPT-4, and Gemini Pro (Anthropic, 2023; OpenAI et al., 2023; Gemini Team et al., 2023); open-weight models such as Llama 3.1, Gemma 2, and Mixtral (Dubey et al., 2024; Gemma Team et al., 2024b; Jiang et al., 2024); and open-source models such as OLMo (Groeneveld et al., 2024). Except for open-source models, where the data is available and thus the language distribution is transparent, there is confusion regarding the capabilities/language distribution of these LLMs in other languages.

Primarily, the focus in evaluating LLMs has been on developing benchmarks to assess their performance in English. Most benchmarks in multilingual setups consist of classical monolingual NLP tasks such as sequence labeling (Ahuja et al., 2023; Lai et al., 2023a), the automatic translation of popular English benchmarks such as MMLU (Hendrycks et al., 2021) into a limited number of languages (Lai et al., 2023b; OpenAI, 2024), or language-specific benchmarks for languages such as Persian (Ghahroodi et al., 2024), Arabic (Koto et al., 2024), Korean (Son et al., 2024), Turkish (Yüksel et al., 2024), and Chinese (Li et al., 2024c).

Most LLMs are English-centric, either by choice or due to the availability of abundant data sources in English. Either way, for these models to be effective in other languages, it is important that the other languages align with the main language, i.e., English. Given such alignment, English could act as a *"rising tide that raises all ships,"* meaning that improvements in English performance could benefit other languages, especially in tasks such as reasoning (Zhu et al., 2024). Contrarily, if a language does not align well with English, an English-centric LLM may not provide *meaningful coverage* for

that language. Indeed, Wendler et al. (2024) have found that for Llama 2 (Touvron et al., 2023b), an English-centric LLM, English could be seen as a kind of "pivot" language, enabling to solve complex semantic tasks in a foreign language through a detour into English. More precisely, they show that Llama 2 was able to decode semantically correct next tokens in the middle layers, assigning higher probabilities to the English tokens than to the foreign version, which is only selected in the upper layers. Zhao et al. (2024) present a hypothesis regarding the middle layers of English-centric LLMs, suggesting that these models use English as a means of reasoning while incorporating multilingual knowledge. Based on their analysis, the number of language-specific neurons in the middle layers decreases within the self-attention mechanism but remains consistent across the layers of the feed-forward structure when processing multilingual queries.

In this paper, we introduce MEXA, a method that uses the observation that English-centric LLMs semantically use English as a kind of pivot language in their middle layers to evaluate the actual multilingual coverage of LLMs. This is done by measuring how well the embeddings of parallel sentences in the middle layers of LLMs for non-English languages are aligned with the embedding of their corresponding English. We extensively verify the MEXA estimation of language coverage for each LLM, using Pearson correlation between estimated and actual scores for various tasks. We use two parallel datasets: FLORES-200 (NLLB Team et al., 2022) and Bible (Mayer & Cysouw, 2014); nine LLMs: Llama family, Gemma family, Mistral, and OLMo; and three tasks: Bele-bele (Bandarkar et al., 2024), m-MMLU, and m-ARC (Lai et al., 2023b). Our results show that MEXA achieves a promising average Pearson correlation of 0.90 with established downstream tasks across nine models and two parallel datasets. In our study on the calculation of MEXA scores, we conduct multiple design analyses to examine the impact of token-level pooling for the embeddings (i.e., using the last token versus a weighted average) and layer-level pooling in computing alignment scores. While MEXA demonstrates a high correlation across most setups, we find that a weighted average based on tokens, combined with mean pooling, yields the best results.

## 2 BACKGROUND AND RELATED WORK

We discuss the distribution of pre-training data in LLMs and multilingual evaluation benchmarks in Appendices A.1 and A.2 while focusing on cross-lingual alignment here. Research in the cross-lingual alignment field either aims to uncover the underlying mechanisms of alignment and assess its impact on models and downstream tasks, or attempts to enhance model performance by enforcing alignment before, during, or after the pre-training phase. Most of these papers have focused on encoder-only models, such as XLM-R (Conneau et al., 2020a) and mBERT (Devlin et al., 2019), among others (Hämmerl et al., 2024). In this work, we focus primarily on decoder-only models.

**Understanding Alignment.** Ye et al. (2023) show that English-centric models such as Llama 1 (Touvron et al., 2023a) not only possess multilingual transfer abilities (after fine-tuning on one source language, they can be applied to other languages) but may even surpass the multilingual transfer abilities of multilingual pre-trained models such as BLOOM (BigScience Workshop et al., 2023). Schäfer et al. (2024) find that GPT-2-style decoder-only models show strong cross-lingual generalization when trained on an imbalanced mix of languages. However, when trained on a balanced language set, they do not observe increased performance compared to monolingual settings. Wendler et al. (2024) perform single-token analysis to demonstrate that English-centered LLMs, such as Llama 2, use English semantically as an internal latent language in the middle layers when handling multilingual queries. Zhong et al. (2024) extend this analysis to multiple tokens, also showing that an LLM dominated by both English and Japanese uses both languages as internal latent languages. Zhao et al. (2024) explore how LLMs handle multilingualism. They hypothesize that LLMs initially interpret the query, converting multilingual inputs into English for task-solving. In the middle layers, the models rely on English with self-attention mechanisms for reasoning, while employing multilingual knowledge through feed-forward structures. In the final layers, LLMs generate responses consistent with the original query language. Li et al. (2024f) and Li et al. (2024b) are even more closely related to ours. Li et al. (2024f) employs absolute cosine similarity values between last token embeddings derived from parallel sentences with English to predict the ranking of language performance across various models. However, as we discuss in Section 3, relying solely on absolute cosine values can be misleading, and as shown in Section 5.3, absolute cosine values are less correlated with downstream tasks than MEXA score. Li et al. (2024b) uses English probing tasks and their automatic translations to construct a multilingual evaluation. While they compare

embedding similarity scores between high- and low-resource languages with corresponding evaluation results, similar to Li et al. (2024f), they do not assess whether these correlations hold across other downstream tasks. In Section 5, we demonstrate that MEXA scores align closely with a broad range of downstream tasks.

**Boosting Alignment.** The idea to enforce alignment in encoder-only models using parallel sentences dates back to (Conneau & Lample, 2019), and has been explored under various guises e.g., using mixed-language sentences and/or bilingual dictionaries Huang et al. (2019); Conneau et al. (2020b); Cao et al. (2020); Kulshreshtha et al. (2020); Efimov et al. (2023); Zhang et al. (2023b). Recently, Li et al. (2024d) improve multilingual alignment by initializing the decoder-only models to generate similar representations of aligned words using contrastive learning and preserves this alignment using a code-switching strategy during pretraining. Liu et al. (2024a) propose a data allocation technique to select a core subset of languages for fine-tuning, better aligning the multilingual capabilities of decoder-only LLMs and making them more truthful in their responses. Li et al. (2024a) propose aligning internal sentence representations across different languages using multilingual contrastive learning and aligning outputs by following cross-lingual instructions in the target language for decoder-only models.

## 3 MEXA

We now describe the MEXA method for computing the alignment score of language $L_1$ with respect to the pivot language $L_2$, given the language model $m$. In this paper, we use the term *cross-lingual alignment*, *geometric alignment*, or simply *alignment* to refer to the semantic similarity of multilingual embeddings across languages. $L_2$, for English-centric LLMs and in this paper, is English. To assess alignment, we use parallel sentences in two languages, $L_1$ and $L_2$.

What defines semantic similarity in multilingual embeddings across languages? The goal of semantic similarity is to ensure that parallel sentences have sufficiently high similarity, reflecting alignment between the two languages. However, considering only the absolute cosine similarity value as the alignment score does not guarantee proper alignment. For some languages, even non-parallel sentences exhibit similarity scores comparable to those of parallel sentences (see §5.3). This is largely due to the high anisotropy observed in Transformer models, which can lead to so-called hubness issues, making it difficult to distinguish between similar and dissimilar embeddings (Ethayarajh, 2019), especially in multilingual models (Hämmerl et al., 2023; Rajaee & Pilehvar, 2022). However, a direct comparative analysis of the cosine similarity between parallel and non-parallel sentence pairs across languages can help overcome these issues. Instead of using the absolute cosine similarity value for alignment, we assign binary values (1 or 0) based on whether a criterion for semantic similarity is satisfied. This criterion imposes that (a) parallel sentences should have high cosine similarity, and (b) non-parallel pairs should also have low similarity values, ensuring the similarity is not random or biased. Specifically, if the cosine similarity for a pair of parallel sentences is higher than for any non-parallel sentences, we assign a value of 1 for this pair; otherwise, a value of 0. This approach sidesteps the hubness problem since the absolute cosine similarity values themselves are not directly used.

To compute MEXA, we first apply the cosine similarity function to the pairs of embeddings of parallel sentences in languages $L_1$ and $L_2$. In Section 3.1, we describe how embeddings can be computed for each layer $l$ of the autoregressive language model $m$. We generate a square matrix $C(L_1, L_2, m, l)$ representing cosine similarities of embeddings at the output of layer $l$ for all parallel sentences in languages $L_1$ and $L_2$. We denote $c_{ij}(l)$ the element in the $i$-th row and $j$-th column of $C(L_1, L_2, m, l)$. It represents the cosine similarity between the $i$-th sentence of $L_1$ and the $j$-th sentence of $L_2$ at layer $l$ of language model $m$. The diagonal elements of $C$, denoted $c_{ii}(l)$, represent the cosine similarity between parallel sentence pairs from $L_1$ and $L_2$. We define the MEXA alignment score ($\mu(.)$) as follows:

$$\mu\big(C(L_1, L_2, m, l)\big) = \frac{1}{n} \sum_{i=1}^{n} \mathbf{1}\left(c_{ii}(l) > \max_{j \in \{1,...,n\} \setminus \{i\}} \{c_{ij}(l), c_{ji}(l)\}\right) \tag{1}$$

where $n$ is the number of diagonal elements (i.e., the dimension of the matrix), and $\mathbf{1}(\cdot)$ is the indicator function, which equals 1 if its argument condition evaluates to true and 0 otherwise. This

alignment score measures how often $c_{ii}(l)$ is the maximum value in both its row and column. The MEXA alignment score can alternatively be understood as a measure of sentence retrieval performance (Hu et al., 2020; Liu et al., 2024b; Hämmerl et al., 2024), with the metric of P@1 applied with queries in language $L_1$ and answers in $L_2$, and vice versa. We discuss other ways to calculate semantic similarity between languages in Appendix A.3.

**Layer-wise Pooling.** The MEXA alignment score $\mu\big(C(L_1, L_2, m, l)\big)$ is computed for language $L_1$ respect to pivot language $L_2$ for each layer $l$ of the language model $m$. To compute a single MEXA alignment score given the language model $m$ and $L_1$, $L_2$, we use mean and max pooling strategies over multiple layers.

### 3.1 SENTENCE EMBEDDINGS

Sentence embeddings are a transformation of a sentence into a fixed-size vector that captures its meaning. The process of computing sentence embeddings can vary depending on the model architecture. Typically, sentence embeddings are used in encoder-based models such as BERT, which employ bidirectional attention. In these models, the hidden states of each token are first extracted, then aggregated, commonly by averaging the hidden states from the output layer. Since attention in these models is bidirectional, each token contributes equally to the final embedding. Alternatively one can use the output of the special [CLS] token as per the original BERT work (Devlin et al., 2019).

In this paper, we focus on autoregressive language models that use a decoder-only architecture. In this architecture, attention is not bidirectional; instead, it takes the form of causal attention (left-to-right). In bidirectional attention, each token has access to every other token in the sequence. However, in causal attention, the embedding of a token at position $t$ is only influenced by the embedding of preceding tokens at positions $0, 1, \ldots, t-1$. Therefore, simple averaging biases the embeddings towards sentence-initial words. Instead, two alternative methods are considered: using only the last token and weighted averaging. We use and compare both methods to acquire the sentence embeddings needed for MEXA.

A standard way to compute a sentence embedding uses only the last token of that sentence. Jiang et al. (2023b) show that using the last token in the format of a prompt template for a sentence $s$, such as 'This sentence: {s} means in one word:', can be effective. Inspired by this, Li & Li (2024) used the prompt 'Summarize sentence {s} in one word:' to obtain the last token embedding as the sentence-level text embedding. However, not all models are instruction-tuned; some earlier works, such as Neelakantan et al. (2022); Wang et al. (2024); Ma et al. (2024), use the last token without any prompt. Since the models studied in this paper are only pre-trained and use multiple languages in the input, we decided to use the last token method without any preceding instruction.

An alternative is weighted averaging. It relies on the intuition that using only the last token might not represent the entire sentence, as the influence of earlier tokens may have diminished. This suggests that tokens at the end of the sentence should contribute more to the overall embedding than those at the beginning. Another motivation is that sentence-final tokens are influenced by preceding tokens and contain more context, while the representation of sentence-initial tokens has significantly less contextual representation. To address this, Muennighoff (2022) proposes to assign weights to each token based on its position. Thus, the sentence embedding of layer $l$ using position-weighted averaging is:

$$e_l = \sum_{t=1}^{T} w_t h_{lt} \quad \text{with} \quad w_t = \frac{t}{\sum_{k=1}^{T} k} \tag{2}$$

where $T$ is the number of tokens in the given sentence, $h_{lt}$ is the embedding of the $t$-th token at layer $l$, and $e_l$ is the sentence embedding at layer $l$.

## 4 EXPERIMENTS

We conduct experiments using various multi-parallel datasets (FLORES-200 and the Bible), models (Llama family, Gemma family, Mistral, and OLMo), and existing benchmarks/tasks (Belebele, m-

MMLU, m-ARC). Our objective is to assess how well the MEXA alignment score from various parallel datasets correlates with the different benchmarks/tasks for different models.

## 4.1 PARALLEL DATA

We calculate the MEXA score using the parallel datasets of FLORES-200 (NLLB Team et al., 2022) and the Bible (Mayer & Cysouw, 2014). While there are other high-quality parallel datasets, such as NTREX-128 (Federmann et al., 2022), IN22 (Gala et al.), OPUS-100 (Zhang et al., 2020), Europarl (Koehn, 2005), OpenSubtitles (Lison & Tiedemann, 2016), TED2020 (Reimers & Gurevych, 2020), and Tatoeba (Tatoeba Community, 2006), we specifically chose FLORES-200 due to its high quality and support for a wide range of languages, while the Bible dataset was selected for its extensive language coverage.

FLORES-200 is a parallel corpus, where the English subset is sourced from Wikimedia projects. Each English sentence has been translated into 204 distinct language-script combinations, these translations have been verified by humans. The dataset contains 997 sentences for development, 1012 sentences for dev-test, and 992 sentences for testing. As the FLORES-200 test set is not publicly accessible, we use the dev-test set as our FLORES parallel test corpus, in line with previous studies. For faster computation, we only consider the first 100 sentences from each language. As shown in Appendix A.4, the odds of the MEXA score randomly achieving a high value with 100 sentences are very slim.

The Parallel Bible (Mayer & Cysouw, 2014) covers a very large number of languages. From this resource, we managed to create a subcorpus, a super parallel dataset of the Bible, with 1,401 language-script labels, each containing 103 sentences (i.e., Bible verses).[1] This corpus includes many low-resource languages, many of which are not covered by existing language technologies (Joshi et al., 2020), and MEXA can be adopted since only parallel data is needed. We use all the 103 sentences from each language.

## 4.2 MODELS

For our experiments, we select models with around 7B parameters, which are considered a base size in the LLM community. The state-of-the-art open-weight models in this range include Llama 1, 2, 3, and 3.1 (Touvron et al., 2023a;b; Meta, 2024; Dubey et al., 2024), Gemma 1 and 2 (Gemma Team et al., 2024a;b), Mistral 0.3 (Jiang et al., 2023a), and the open-source model OLMo 1.7 (Groeneveld et al., 2024). We also select a larger model, Llama 3.1 70B, to show that our findings hold even when scaling further. To apply MEXA, we need to access model weights to compute input sentence embeddings for each layer. We use three popular open-weight model families: Llama, Gemma, and Mistral. As a less multilingual version of state-of-the-art LLMs, we include OLMo, which is trained on a more English-centric corpus of Dolma (Soldaini et al., 2024). Although there are multilingual models such as PolyLM (with support of 18 languages), XGLM (Lin et al., 2022) (with support of 30 languages) and BLOOM (BigScience Workshop et al., 2023) (with support of 46 languages) our focus here is on LLMs which are state-of-the-art in English based tasks such as MMLU (Stanford CRFM, 2024).

## 4.3 BENCHMARKS

Among the existing evaluation benchmarks in Table 4 from Appendix A.2, we chose the Belebele benchmark (Bandarkar et al., 2024), m-ARC (Lai et al., 2023b), and m-MMLU (Lai et al., 2023b), which support the highest number of high-, medium-, and low-resource languages and are directly related to natural understanding tasks, which is the primary focus of this paper.

Belebele is a multiple-choice reading comprehension task designed to assess language models across a range of high-, medium-, and low-resource languages. Each question in the dataset is accompanied by four possible answers and is linked to a brief passage from the FLORES-200 dataset (NLLB Team et al., 2022). The human annotation process was carefully curated to generate questions that effectively differentiate between various levels of language comprehension, supported by rigorous quality checks. Belebele supports 122 distinct labels (language-script combinations) corresponding

---

[1][anonymized url]

| | | Gemma 2 9B | Gemma 1 7B | Llama 3.1 70B | Llama 3.1 8B | Llama 3 8B | Llama 2 7B | Llama 1 7B | Mistral 0.3 7B | OLMo 1.7 7B | AVG |
|---|---|---|---|---|---|---|---|---|---|---|---|
| Task$_{\{eng\}}$ | Belebele | 0.9178 | 0.8467 | **0.9456** | 0.8767 | 0.8689 | 0.4822 | 0.4156 | 0.8389 | 0.7711 | 0.7737 |
| | m-MMLU | 0.6998 | 0.6138 | **0.7700** | 0.6315 | 0.6294 | 0.4523 | 0.3569 | 0.5988 | 0.5210 | 0.5859 |
| | m-ARC | 0.6775 | 0.5870 | **0.7014** | 0.5794 | 0.5836 | 0.5128 | 0.5000 | 0.5862 | 0.4872 | 0.5795 |
| Task$_{L\setminus\{eng\}}$ | Belebele (avg., \|L\| = 116) | 0.7093 | 0.5633 | **0.7684** | 0.5705 | 0.5533 | 0.3028 | 0.2755 | 0.4457 | 0.3627 | 0.5057 |
| | m-MMLU (avg., \|L\| = 33) | 0.5582 | 0.4734 | **0.6384** | 0.4720 | 0.4664 | 0.3260 | 0.2807 | 0.4207 | 0.3390 | 0.4416 |
| | m-ARC (avg., \|L\| = 31) | 0.4779 | 0.4220 | **0.5054** | 0.3941 | 0.3892 | 0.3174 | 0.2970 | 0.3662 | 0.2731 | 0.3825 |
| FLORES | $\mu_{\text{Mean}}$ (avg., \|L\| = 116) | **0.5088** | 0.3815 | 0.4110 | 0.3963 | 0.3939 | 0.0866 | 0.1946 | 0.2642 | 0.0413 | 0.2976 |
| | $\mu_{\text{Max}}$ (avg., \|L\| = 116) | 0.7194 | 0.5872 | **0.7725** | 0.6538 | 0.6520 | 0.2464 | 0.3579 | 0.4716 | 0.1965 | 0.5175 |
| Bible | $\mu_{\text{Mean}}$ (avg., \|L\| = 101) | **0.3568** | 0.2152 | 0.3169 | 0.2103 | 0.2026 | 0.1246 | 0.0908 | 0.1198 | 0.0121 | 0.1832 |
| | $\mu_{\text{Max}}$ (avg., \|L\| = 101) | 0.6076 | 0.4021 | **0.6599** | 0.4212 | 0.4190 | 0.2724 | 0.2357 | 0.2606 | 0.0319 | 0.3678 |

Table 1: $\mu_{\text{pooling}}$ indicates the MEXA score for each corresponding pooling strategy. The embeddings are computed using weighted average based on token positions (Eq. 2). Top values are in **bold**, with second-best underlined.

to 115 distinct languages. However, FLORES-200 does not support 5 of these labels, corresponding to Romanized versions of 5 Indic languages. Therefore, we conducted our analysis between the FLORES-200 and Belebele benchmarks on 117 common labels. Additionally, there are 102 common labels between the Bible parallel data and Belebele benchmark.

Both ARC Challenge (Clark et al., 2018) and MMLU (Hendrycks et al., 2021) are also set up as multiple-choice question-answering tasks, but they focus on different types of knowledge and reasoning skills. ARC Challenge is classified as a common-sense reasoning task, consisting of grade-school level science questions, while MMLU consists of questions across a wide range of subjects, including humanities, social sciences, and more. Lai et al. (2023b) used GPT-3.5-turbo (OpenAI, 2022) and a translation prompt to translate examples from both datasets and create m-ARC and m-MMLU in 31 languages (excluding English). Later, m-MMLU was expanded to also include Icelandic (isl_Latn) and Norwegian (nob_Latn). The Icelandic portion was translated using the Mideind.is, while Norwegian was generated with DeepL.com.[2] m-MMLU consists of 277 questions in its training set, 13,258 in the test set and 1,433 in the validation set. m-ARC consists of 1,116 questions in the training set, 1,169 in the test set, and 298 in the validation set. We use the entire test set for each of these benchmarks to evaluate the models, except in one case. For Llama 3.1 70B, we use the first 500 questions of m-MMLU instead of the whole set due to resource constraints. Since the selected LLMs used in our experiment are not instruction-tuned, we use 5-shot in-context learning with the lm-evaluation-harness framework, employing log-likelihood-based multiple-choice scoring. Other settings, such as prompt templates, are configured according to the framework's default (Gao et al., 2023; Biderman et al., 2024).

### 4.4 EVALUATION MEASURES

We use Pearson correlation to assess the strength of the correlation between MEXA and downstream performance on our evaluation benchmarks. Pearson correlation is a statistical measure that calculates the strength and direction of the linear relationship between two variables. A high Pearson correlation would indicate that MEXA provides a reliable assessment of multilingual capabilities in English-centric LLMs.

## 5 RESULTS

Table 1 presents the downstream performance of the selected models across three benchmarks, along with MEXA scores from two parallel datasets. Notably, among models with parameter sizes ranging from 7 to 9 billion, both Gemma 2 and Llama 3.1 outperform the others in terms of non-English downstream performance and MEXA scores. The Llama 3.1 and Llama 3 models exhibit similar alignment and downstream task performance; yet, both represent substantial advancements compared to Llama 2. Moreover, results for the Llama 3.1-70B model indicate that scaling can significantly enhance alignment when compared to its smaller version. Interestingly, while Mistral achieves comparable results to Gemma 1 on English benchmarks, it demonstrates inferior alignment, which likely accounts for its reduced performance on non-English tasks. Furthermore, the Llama 2 model achieves higher MEXA scores than OLMo, indicating better alignment. However,

---

[2]https://huggingface.co/datasets/alexandrainst/m_mmlu

| | | Gemma 2 9B | Gemma 1 7B | Llama 3.1 70B | Llama 3.1 8B | Llama 3 8B | Llama 2 7B | Llama 1 7B | Mistral 0.3 7B | OLMo 1.7 7B | AVG |
|---|---|---|---|---|---|---|---|---|---|---|---|
| **FLORES** — weighted average | $\rho\,(\mu_{\text{Mean}}, \text{Belebele})$ | 0.9247 | 0.9421 | 0.8291 | 0.9478 | 0.9588 | 0.8364 | 0.8404 | 0.9732 | 0.8425 | 0.8994 |
| | $\rho\,(\mu_{\text{Max}}, \text{Belebele})$ | 0.9623 | 0.9676 | 0.9211 | 0.9392 | 0.9326 | 0.8362 | 0.7649 | 0.9448 | 0.9198 | 0.9098 |
| | $\rho\,(\mu_{\text{Mean}}, \text{m-MMLU})$ | 0.9342 | 0.9697 | 0.9362 | 0.9689 | 0.9647 | 0.9223 | 0.9406 | 0.9857 | 0.9393 | 0.9513 |
| | $\rho\,(\mu_{\text{Max}}, \text{m-MMLU})$ | 0.9060 | 0.9596 | 0.8946 | 0.9003 | 0.8892 | 0.9386 | 0.8936 | 0.9311 | 0.9565 | 0.9188 |
| | $\rho\,(\mu_{\text{Mean}}, \text{m-ARC})$ | 0.9741 | 0.9706 | 0.9374 | 0.9515 | 0.9562 | 0.9052 | 0.9268 | 0.9693 | 0.8630 | **0.9393** |
| | $\rho\,(\mu_{\text{Max}}, \text{m-ARC})$ | 0.9187 | 0.9499 | 0.8736 | 0.8582 | 0.8663 | 0.9297 | 0.8439 | 0.9001 | 0.8298 | 0.8856 |
| **FLORES** — last token | $\rho\,(\mu_{\text{Mean}}, \text{Belebele})$ | 0.8997 | 0.9326 | 0.8491 | 0.9494 | 0.9581 | 0.9141 | 0.8340 | 0.9679 | 0.9467 | **0.9168** |
| | $\rho\,(\mu_{\text{Max}}, \text{Belebele})$ | 0.9225 | 0.9309 | 0.9127 | 0.9244 | 0.9123 | 0.9125 | 0.7693 | 0.9460 | 0.9218 | 0.9058 |
| | $\rho\,(\mu_{\text{Mean}}, \text{m-MMLU})$ | 0.9086 | 0.9637 | 0.9370 | 0.9687 | 0.9690 | 0.9771 | 0.9301 | 0.9659 | 0.9700 | **0.9545** |
| | $\rho\,(\mu_{\text{Max}}, \text{m-MMLU})$ | 0.8448 | 0.9297 | 0.8645 | 0.9224 | 0.9177 | 0.9699 | 0.8902 | 0.9161 | 0.9649 | 0.9134 |
| | $\rho\,(\mu_{\text{Mean}}, \text{m-ARC})$ | 0.9190 | 0.9541 | 0.9524 | 0.9536 | 0.9617 | 0.9390 | 0.9146 | 0.9451 | 0.7356 | 0.9195 |
| | $\rho\,(\mu_{\text{Max}}, \text{m-ARC})$ | 0.8569 | 0.9147 | 0.9005 | 0.8944 | 0.8879 | 0.9464 | 0.8263 | 0.8859 | 0.7037 | 0.8685 |
| **Bible** — weighted average | $\rho\,(\mu_{\text{Mean}}, \text{Belebele})$ | 0.8360 | 0.8530 | 0.7909 | 0.8781 | 0.8974 | 0.8982 | 0.8404 | 0.9118 | 0.7410 | 0.8496 |
| | $\rho\,(\mu_{\text{Max}}, \text{Belebele})$ | 0.8863 | 0.9001 | 0.8851 | 0.9242 | 0.9302 | 0.8926 | 0.8230 | 0.9337 | 0.7549 | **0.8811** |
| | $\rho\,(\mu_{\text{Mean}}, \text{m-MMLU})$ | 0.8051 | 0.8886 | 0.8958 | 0.9096 | 0.8964 | 0.9252 | 0.9159 | 0.9093 | 0.7944 | **0.8823** |
| | $\rho\,(\mu_{\text{Max}}, \text{m-MMLU})$ | 0.5501 | 0.8831 | 0.7748 | 0.8683 | 0.8364 | 0.9180 | 0.9085 | 0.9107 | 0.7388 | 0.8210 |
| | $\rho\,(\mu_{\text{Mean}}, \text{m-ARC})$ | 0.8505 | 0.8998 | 0.9188 | 0.9267 | 0.9116 | 0.8940 | 0.9208 | 0.9317 | 0.8623 | **0.9018** |
| | $\rho\,(\mu_{\text{Max}}, \text{m-ARC})$ | 0.6070 | 0.8803 | 0.8030 | 0.8769 | 0.8552 | 0.8684 | 0.8879 | 0.9178 | 0.8220 | 0.8354 |
| **Bible** — last token | $\rho\,(\mu_{\text{Mean}}, \text{Belebele})$ | 0.7656 | 0.8005 | 0.5944 | 0.7934 | 0.8396 | 0.9046 | 0.8299 | 0.9177 | 0.8866 | 0.8147 |
| | $\rho\,(\mu_{\text{Max}}, \text{Belebele})$ | 0.7844 | 0.8299 | 0.5264 | 0.8000 | 0.8100 | 0.9047 | 0.8048 | 0.9235 | 0.8796 | 0.8070 |
| | $\rho\,(\mu_{\text{Mean}}, \text{m-MMLU})$ | 0.7194 | 0.7646 | 0.6472 | 0.6068 | 0.6516 | 0.8827 | 0.8692 | 0.8672 | 0.8060 | 0.7572 |
| | $\rho\,(\mu_{\text{Max}}, \text{m-MMLU})$ | 0.7075 | 0.6886 | 0.5037** | 0.5228** | 0.4461** | 0.9079 | 0.8576 | 0.8643 | 0.7994 | 0.6998 |
| | $\rho\,(\mu_{\text{Mean}}, \text{m-ARC})$ | 0.7411 | 0.7754 | 0.6592 | 0.5976 | 0.6494 | 0.8537 | 0.8537 | 0.8927 | 0.6997 | 0.7469 |
| | $\rho\,(\mu_{\text{Max}}, \text{m-ARC})$ | 0.7293 | 0.7000 | 0.5190** | 0.5335** | 0.4853** | 0.8494 | 0.8309 | 0.8624 | 0.6867 | 0.6885 |

Table 2: Pearson correlation of MEXA using FLORES and Bible data across three tasks. $\rho\,(\mu_{\text{Pooling}}, \text{Task})$ is the correlation of MEXA for the corresponding pooling strategy and benchmark. In all settings except **, the p-value is $p < 0.001$. The best average correlations for each task are in **bold**, and the second bests are underlined.

due to Llama 2's weaker performance on English tasks, it fails to transfer this alignment effectively, leading to comparable non-English performance between Llama 2 and OLMo. This observation is further explored in Section 5.2, where we normalize the expected performance based on the pivot language, namely English.

## 5.1 MEXA CORRELATION ANALYSIS

We compute sentence embeddings for the selected models using two methods: weighted average based on token positions and last token (see §3.1). We apply mean and max pooling on the MEXA alignment scores across all model layers to derive a single score for each language. In Table 2, we report the correlation between the MEXA scores (computed using both mean- and max-pooling, for the two embedding methods) and task performances. Across all settings, the best overall result (higher correlation) is achieved when embeddings are computed using the **weighted average**, with **mean pooling** as the pooling method. We adopt this configuration as the default setting for MEXA.

**FLORES vs Bible.** In the default setting, the average Pearson correlation score for the FLORES parallel dataset across different tasks is 0.9300, while for the Bible parallel dataset, it is 0.8779. The reason the Bible scores are generally lower than FLORES is that FLORES data is cleaner and more aligned with modern, standardized texts, whereas the Bible data is older and more specialized. For example for some languages, the orthography of Bible texts no longer matches today's orthography. In the Bible, Arabic often includes diacritics, which are typically omitted in modern writing and tasks, making the text less familiar to models trained on contemporary data. Additionally, although the Bible dataset has been made parallel, sentence alignment can still be inconsistent due to translation nuances. In contrast, FLORES is carefully curated to ensure high-quality, sentence-level parallelism across languages for machine translation tasks.

**Weighted Average vs. Last Token Embeddings.** The use of last token embeddings shows promisingly high correlations with the FLORES parallel data; however, for the Bible dataset, the correlation is low in some cases. We believe this may stem from the high occurrence of Bible sentences (especially in English), which leads models to memorize these phrases. Using the WIMBD toolkit (Elazar et al., 2024), we found that, on average, there are 92 times more documents in Dolma 1.7 (Soldaini et al., 2024) containing exact Bible sentences than those in FLORES. Consequently, when using Bible examples, the last token is biased towards predicting the specific memorized next token rather than incorporating context-related signals. Therefore, one should consider the hazard of memorized data when using last token embeddings. The weighted-average method, which takes into account

the influence of multiple tokens, can mitigate the impact of a poor embedding for the last token and enable the model to capture useful information from the other tokens more robustly.

**Max Pooling vs. Mean Pooling.** In our comparison of mean pooling and max pooling on the Belebele benchmark, we found that mean pooling underestimates low-resource languages (resulting in more MEXA scores near 0), while max pooling correlates better with the Belebele benchmark. This can be explained by the fact that Belebele is an easier task among the three evaluated, allowing even low-resource languages to achieve good scores. Conversely, based on our experiment with m-ARC, max pooling tends to overestimate low-resource languages, making mean pooling more aligned with m-ARC. This can be attributed to m-ARC being the most challenging task among the three, where even medium-resource languages do not achieve high scores. Changing the pooling method from mean to max can be considered when dealing with different levels of understanding.

## 5.2 DOWNSTREAM PERFORMANCE ESTIMATION

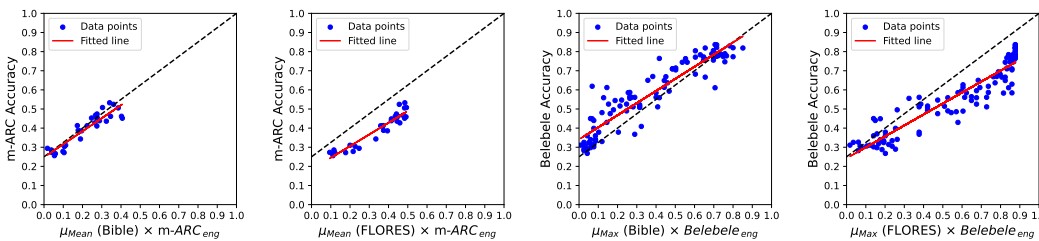

Figure 1: The relationship between MEXA scores of Llama 3.1-8B from the Bible and FLORES, adjusted by the English task performance, for tasks in Belebele and the m-ARC benchmark.

A complete Pearson correlation (i.e. $\rho = 1.0$) indicates that a linear equation perfectly describes the relationship between MEXA and the evaluation benchmarks, with all data points lying on a line. Given the high correlation values shown in Table 2, it is reasonable to conclude that we can fit a line that closely approximates this linear relationship. This line converts the MEXA scores back to downstream task performances. We employed a linear model to predict this line by minimizing the residual sum of squares between the MEXA scores (multiplied by the performance on the English task) and the task performances. We needed to adjust the MEXA scores for this purpose, as the MEXA score for language $L_1$ indicates how well $L_1$ is aligned with English but does not reflect the estimated task performance of the model for language $L_1$. Of course, this does not change the Pearson correlation, as it is unaffected by linear transformations. The three tasks considered in this paper involve multiple-choice questions with four possible answers for each question, resulting in a chance of being randomly correct of $\frac{1}{4}$. However, the minimum score for MEXA scores is 0. Thus, the ideal slope for the line would be $\frac{3}{4}$ with an intercept of $\frac{1}{4}$ (X-axis: adjusted MEXA scores, Y-axis: task performance). In Figure 1, we plot this relationship for Llama 3.1-8B using the Bible and FLORES parallel datasets for Belebele and m-ARC. We chose mean pooling for Belebele and max pooling for m-ARC, since these pooling methods yield a stronger correlation (see §5.1). The pairs of (slope, intercept) from left to right in the Figure 1 are: (0.6804, 0.2477), (0.6103, 0.1838), (0.6340, 0.3408) and (0.5726, 0.2423). With data points from both high-resource and low-resource languages, this line can be calculated; otherwise, the ideal line may be used as a reference.

**Language Coverage.** We present the adjusted MEXA score for all languages available in FLORES-200 in Table 5 from Appendix A.5 for the selected models. The languages are categorized into groups ranging from well-covered to not covered. In Table 5, we can clearly see that Llama 3.1-70B and Gemma 2-9B show a higher level of multilinguality than other models.

## 5.3 MEXA VS ABSOLUTE COSINE SIMILARITY

We compare MEXA with the use of absolute cosine similarities. To begin with, cosine similarity scores are not always directly comparable across models. For example, if a language shows a higher cosine similarity with English for one model than another, it does not necessarily indicate better alignment in the former model. However, MEXA has the advantage of being directly comparable, as

its score does not rely on absolute similarity values. To examine the correlation of both methods with downstream tasks, we conducted the following experiment. We used parallel data from FLORES and downstream task data from the Belebele benchmark, focusing on 116 common labels. For each non-English language, we computed the average absolute cosine similarity for parallel sentences with English, and the average absolute cosine similarity for non-parallel sentences with English. Following the setup by Li et al. (2024f), which employs absolute cosine similarity values to predict the performance and rank of languages, we computed the embeddings using the last-token method and applied mean pooling over layers {5, 10, 15, 20, 25}. We report results using the Gemma 1 and Llama 1 7B models, which are commonly used in our experiments. For a fair comparison, this setup is applied to both absolute cosine similarity and MEXA. For the Gemma 1 model, MEXA achieves a correlation of 0.9260 with downstream task performance, while the absolute cosine similarity for parallel sentences achieves a correlation of 0.7651. Additionally, the correlation between the absolute cosine similarity for parallel and non-parallel sentences is 0.9232. For the Llama 1 model, MEXA achieves a correlation of 0.8365 with downstream task performance, while the absolute cosine similarity for parallel sentences achieves a correlation of 0.6473. Additionally, the correlation between the absolute cosine similarity for parallel and non-parallel sentences is 0.9064. In both models, the absolute cosine similarity method achieved significantly lower correlations with downstream tasks compared to MEXA. This discrepancy arises primarily because, for some languages, the similarity score can be high regardless of whether the sentences are parallel or non-parallel. Furthermore, a low similarity score between two languages does not necessarily indicate weak alignment, as overall similarity scores may be low while parallel sentences still exhibit much higher scores than non-parallel ones.

## 5.4 VISUALIZATION OF LAYERS

In Figure 2, we show the results of applying MEXA to 20 pairs of language_script from FLORES parallel dataset for Llama 1-7B and Llama 3.1-8B across all 32 layers. We selected these languages from different families, writing systems, and both high- and low-resource categories. The embeddings are computed using weighted average based on token positions. Figure 2 shows that high-resource languages (with more prevalence on the web; see §A.1) achieve higher alignment scores across different layers, while low-resource languages achieve lower scores. In the initial layers, embeddings are more in-language, resulting in lower alignment scores. As embeddings progress to the mid-layers, they become more aligned with the main language of the LLM, i.e., English.

MEXA is comparable between models as long as the same parallel dataset and setting is used to obtain the MEXA scores. Figure 2 shows that in many languages, particularly high-resource languages, Llama 3.1 achieves a significantly higher alignment score than its predecessor, Llama 1. Although Llama 3.1 exhibits better alignment scores with English for medium and low-resource languages, there is still significant room for improvement. Comparing Arabic (arb_Arab) with its romanized version (arb_Latn), we see that both Llama 1 and Llama 3.1 models perform better in the native script than in the Latin script, even though Llama 1's tokenizer for Arabic is essentially a character-based tokenizer. In general, for very low-resource languages, those in Latin script tend to have higher alignment scores, likely because the tokenization is more favorable for Latin characters.

In Figure 3, we display the t-SNE (Van der Maaten & Hinton, 2008) plots of the embeddings of Figure 2 from 3 different layers of Llama 3.1: embedding layer 0, mid-layer 13, and last layer 32. We assign a different color to each language. For layers 0 and 32, the embeddings are more language-specific, while in the mid-layer, they become more language-neutral. Languages that maintain their language-specific embeddings in the mid-layer are clustered separately and, notably, correspond to the very low-resource languages that receive the worst alignment scores from MEXA.

## 6 CONCLUSION

We introduce MEXA, a method for assessing the multilingual capabilities of English-centric large language models (LLMs). MEXA builds on the observation that English-centric LLMs semantically use English as a pivot language in their intermediate layers. MEXA computes the alignment between non-English languages and English using parallel sentences, estimating the transfer of language understanding capabilities from English to other languages through this alignment. This metric can be useful in estimating task performance, provided we know the English performance in the task and the

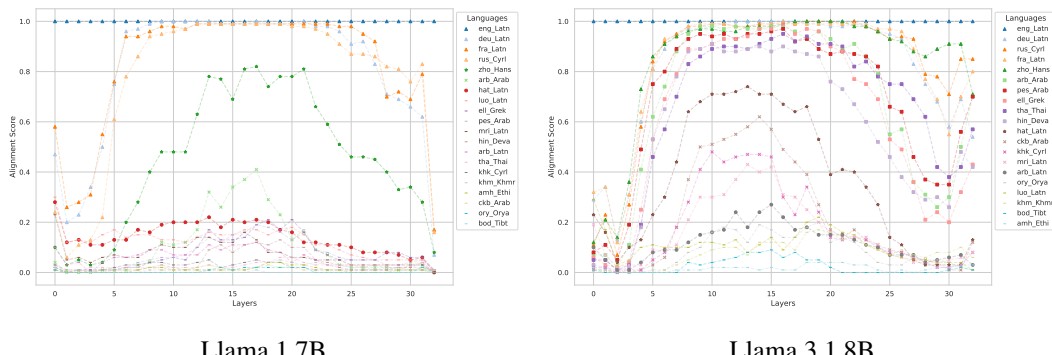

Llama 1 7B                    Llama 3.1 8B

Figure 2: Llama 1 vs. Llama 3.1 agreement score for different languages across all layers. Best performance markers in order: △, □, ⋆, ×, ◦, ₋

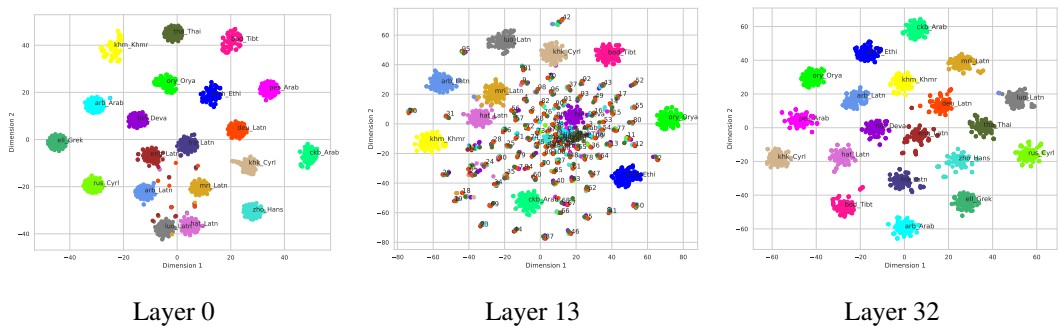

Layer 0                    Layer 13                    Layer 32

Figure 3: Llama 3.1 t-SNE plots for 3 different layers. As shown, in the mid-layers, the embeddings become more language-neutral. The numbers shown in the mid-layers are the IDs of English sentences that are scattered.

alignment score between languages derived from a parallel dataset. Through different studies with two parallel datasets (FLORES-200 and the Bible); a diverse range of LLMs including the Llama family, Gemma family, Mistral, and OLMo, and three downstream tasks (Belebele, m-MMLU, and m-ARC), we demonstrated that MEXA provides a reliable estimation of multilingual performance. For MEXA score calculations, multiple design analyses are conducted to explore the impact of token-level pooling for embeddings and layer-level pooling in computing alignment scores. While MEXA shows high correlation across most configurations, a weighted average of tokens combined with mean pooling delivers the best results. The results reveal a promising average Pearson correlation of 0.90 with established downstream tasks across nine models and two parallel dataset. Overall, MEXA proves to be a valuable method for practitioners aiming to assess the multilingual capabilities of English-centric LLMs, paving the way for future efforts to expand these models to a wider range of underrepresented languages.

# 7 LIMITATION

MEXA provides a rough estimate of the multilingual capabilities of pre-trained English-centric LLMs. Different tasks offer diverse perspectives on the abilities of LLMs, and MEXA cannot replace all of them. Our goal is to highlight the multilingual potential of English-centric LLMs and propose a simple way to evaluate them. We hope this encourages the development of more multilingual LLMs, even though they are likely to contain large shares of English data. Additionally, it is important to note that answers across languages do not always need to be fully aligned (Naous et al., 2024), and for such cases, language- and culture-specific evaluation benchmarks should be developed.

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

## A APPENDIX

### A.1 DISTRIBUTION OF PRE-TRAINING DATA IN LLMS

The distribution of languages in the training data of state-of-the-art LLMs is rarely fully documented. Llama 2 (Touvron et al., 2023b) is a counter-example and its authors have disclosed the language distribution use in pretraining. Their analysis uses the FastText (Bojanowski et al., 2017) language identification tool and a threshold of $0.5$ for the language detection. We reproduce Touvron et al. (2023b, Table 10), which lists 27 languages with percentages greater than 0.005% in the Llama 2 pre-training data, in Table 3. English, with 89.70%, constitutes the vast majority of the training data.

All the languages listed in Table 3 have a presence of more than 0.10% (top 35 languages) on the web according to the W3Techs report (W3Techs, 2024) or more than 0.15% (top 36 languages) according to CommonCrawl (first three snapshots of 2024) (Common Crawl, 2024). However, not all of the most prevalent languages on the web appear in Table 3. The following 9 languages are missing, most of which use non-Latin writing systems: Turkish (tur_Latn), Persian (pes_Arab), Arabic (ara_Arab), Greek (ell_Grek), Hebrew (heb_Hebr), Thai (tha_Thai), Hindi (hin_Deva), Slovak (slk_Latn), and Lithuanian (lit_Latn).

The distribution of data in the training of English-centric LLMs is not the same as on the web, but it does have some correlation. The amount of English in LLM pre-training data is significantly larger than for other languages. This is also observable for GPT-3 (Brown et al., 2020a), where more than 92% of the training texts was in English (Brown et al., 2020b). The rest of the top languages in the data of such models are mostly high-resource languages, which have the most available data on the web (top 36 languages). However, in some models, this could be adjusted by design, for example, to make writing systems with non-Latin languages less prominent (as seen in Llama 2). This weakens the correlation between LLMs' pretraining data and the web.

| Language | Common Script | Percent | Language | Common Script | Percent |
|---|---|---|---|---|---|
| English (eng) | Latn | 89.70% | Ukrainian (ukr) | Cyrl | 0.07% |
| Unknown (unk) | - | 8.38% | Korean (kor) | Hang | 0.06% |
| German (deu) | Latn | 0.17% | Catalan (cat) | Latn | 0.04% |
| French (fra) | Latn | 0.16% | Serbian (srp) | Cyrl/Latn | 0.04% |
| Swedish (swe) | Latn | 0.15% | Indonesian (ind) | Latn | 0.03% |
| Chinese (zho) | Hans/Hant | 0.13% | Czech (ces) | Latn | 0.03% |
| Spanish (spa) | Latn | 0.13% | Finnish (fin) | Latn | 0.03% |
| Russian (rus) | Cyrl | 0.13% | Hungarian (hun) | Latn | 0.03% |
| Dutch (nld) | Latn | 0.12% | Norwegian (nor) | Latn | 0.03% |
| Italian (ita) | Latn | 0.11% | Romanian (ron) | Latn | 0.03% |
| Japanese (jpn) | Jpan | 0.10% | Bulgarian (bul) | Cyrl | 0.02% |
| Polish (pol) | Latn | 0.09% | Danish (dan) | Latn | 0.02% |
| Portuguese (por) | Latn | 0.09% | Slovenian (slv) | Latn | 0.01% |
| Vietnamese (vie) | Latn | 0.08% | Croatian (hrv) | Latn | 0.01% |

Table 3: Language distribution in the pretraining data for Llama 2. The large "Unknown" category is partially composed of programming code data. Common scripts are sourced from the GlotScript resource (Kargaran et al., 2024).

### A.2 MULTILINGUAL EVALUATION BENCHMARKS

Multilingual evaluation methods and the development of benchmarks not only facilitate the assessment of diverse language representations in LLMs but also help in monitoring cross-lingual generalization, to assess the effect of quantization across multiple languages (Marchisio et al., 2024), the development of language-specific models (Tejaswi et al., 2024), and the optimization of safety preferences (Li et al., 2024e), among others. In Table 4, we list benchmarks with the largest language coverage. This list includes benchmarks referenced by MEGA (Ahuja et al., 2023), MEGA-VERSE (Ahuja et al., 2024), xP3 (Muennighoff et al., 2023), the Aya collection (Singh et al., 2024), the lm-evaluation-harness framework (Gao et al., 2023; Biderman et al., 2024), and inter alia. These datasets comprise a mix of translated datasets, some human-translated or verified by native speakers such as AfriXNLI (Adelani et al., 2024) and some relying only on machine translation Lai et al. (2023b). Additionally, there are datasets created independently for each language, such as XLSum (Hasan et al., 2021), where the data is not parallel and the size of the data varies between languages.

Despite the efforts reflected in Table 4, the community is still lacking highly multilingual benchmarks for tasks such as natural language understanding or text generation.

| Dataset | Task | # Languages |
|---|---|---|
| XNLI (Conneau et al., 2018) | Natural Language Inference | 15 |
| IndicXNLI (Aggarwal et al., 2022) | Natural Language Inference | 11 |
| AfriXNLI (Adelani et al., 2024) | Natural Language Inference | 15 |
| m_HellaSwag (Lai et al., 2023b) | Natural Language Inference | 31 |
| PAWS-X (Yang et al., 2019) | Paraphrase Identification | 7 |
| XCOPA (Ponti et al., 2020) | Commonsense Reasoning | 11 |
| XStoryCloze (Lin et al., 2022) | Commonsense Reasoning | 11 |
| m-ARC (Lai et al., 2023b) | Common Sense Reasoning | 31 |
| TyDiQA (Clark et al., 2020) | Question Answering | 11 |
| MLQA (Lewis et al., 2020) | Question Answering | 7 |
| XQuAD (Artetxe et al., 2020) | Question Answering | 11 |
| IndicQA (Doddapaneni et al., 2023) | Question Answering | 10 |
| AfriQA (Ogundepo et al., 2023) | Question Answering | 10 |
| m_TruthfulQA (Lai et al., 2023b) | Multiple Choice Question Answering | 31 |
| UDPOS 2.7 (de Marneffe et al., 2021) | Part of Speech Tagging | 104 |
| WikiANN (Pan et al., 2017) | Name Entity Recognition | 282 |
| XLSum (Hasan et al., 2021) | Summarization | 44 |
| WikiLingua (Ladhak et al., 2020) | Summarization | 18 |
| Belebele (Bandarkar et al., 2024) | Multiple Choice Reading Comprehension | 115 |
| AfriMMLU (Adelani et al., 2024) | Multiple Choice Knowledge Question Answering | 17 |
| m-MMLU (Lai et al., 2023b) | Multiple Choice Knowledge Question Answering | 31 |
| MMMLU (OpenAI, 2024) | Multiple Choice Knowledge Question Answering | 15 |
| M3Exam (Zhang et al., 2023a) | Multimodal Multiple Choice Knowledge Question Answering | 9 |

Table 4: Multilingual evaluation benchmarks

### A.3 SEMANTIC SIMILARITY IN MULTILINGUAL EMBEDDINGS

There are other ways to compute similarity between languages, such as Representational Similarity Analysis (RSA) (Chrupała & Alishahi, 2019) and Central Kernel Alignment (CKA) (Kornblith et al., 2019). RSA involves first computing the cosine similarity for sentence embeddings within each language, then correlating these in-language similarities with those in other languages. CKA, another metric, is adopted by Conneau et al. (2020b) and Muller et al. (2021). Conneau et al. (2020b) show that the CKA similarity is highly correlated with sentence retrieval scores for four languages. In this paper, our focus is not on finding different ways to calculate similarity between languages, but on how helpful a properly defined alignment score can be in estimating the multilingual capabilities of LLMs across multiple languages.

### A.4 ROBUSTNESS OF MEXA

We show that the MEXA alignment score ($\mu(.)$) is very robust, and the odds of this score randomly achieving a high value are very slim. Recall that $\mu\big(C(L_1, L_2, m, l)\big)$ measures the fraction of diagonal elements in matrix $C(L_1, L_2, m, l)$ that have the maximum value in their respective rows and columns. If this condition is met $k$ times out of $n$ diagonal elements, then $\mu\big(C(L_1, L_2, m, l)\big)$ is $\frac{k}{n}$. In an $n \times n$ random matrix, the probability of a diagonal element being the maximum in its row and column (a total of $2n-1$ elements) is $p = \frac{1}{2n-1}$. The probability that at least $k$ out of $n$ independent variables are satisfied, given that the diagonal element is the maximum in its row and column, can be computed using the binomial distribution with Eq. 3.

$$P(X \geq \frac{k}{n}) = 1 - \sum_{i=0}^{k-1} \binom{n}{i} p^i (1-p)^{n-i} \tag{3}$$

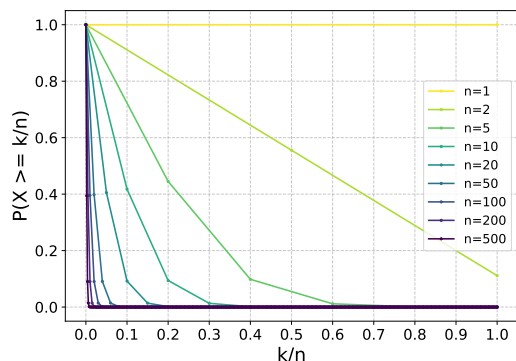

Figure 4: The probability that at least $k$ out of $n$ diagonal elements in an $n \times n$ random matrix are the maximum elements in their respective rows and columns.

In Figure 4, we plot $P(X \geq \frac{k}{n})$. This plot illustrates that, given a sufficient number of parallel sentences ($n$), the probability of achieving a high score by chance is very low. For example, with $n = 100$, the chance of obtaining MEXA alignment score larger than 0.05 ($k = 5$) from a $100 \times 100$ random matrix is $P(X \geq 0.05) = 0.00016$.

## A.5 MEXA FOR FLORES-200

We compute MEXA with weighted average embedding and max pooling for the FLORES parallel data for 203 langauge labels, multiplied by the performance of Belebele for each model in English. We show the results in Table 5, and color the cells based on 0.2 intervals from green (well-covered) to red (not covered): (1.0-0.8), (0.8-0.6), (0.6-0.4), (0.4-0.2), (0.2-0). Note that although FLORES is a high-quality, human-translated dataset, we addressed two major issues before proceeding, as noted by Kargaran et al. (2023). First, the data labeled as Cantonese (Yue Chinese) is not actually Cantonese, so we removed it. Second, the code for Central Atlas Tamazight (tzm), which actually refers to Standard Moroccan Tamazight (zgh), was renamed accordingly. As Belebele is relatively an easy task since the models get good scores in English, and we are using max pooling, this gives a high estimate of the coverage the LLMs have. If the score for a language is not very high, it likely indicates that for more challenging tasks, it will remain low. In Table 5, we can clearly see that Llama 3.1-70B and Gemma 2-9B show a higher level of multilinguality than other models.

| | Gemma 2 9B | Gemma 1 7B | Llama 3.1 70B | Llama 3.1 8B | Llama 3 8B | Llama 2 7B | Llama 1 7B | Mistral 7B | OLMo 7B | AVG |
|---|---|---|---|---|---|---|---|---|---|---|
| eng_Latn | 0.92 | 0.85 | 0.95 | 0.88 | 0.87 | 0.48 | 0.42 | 0.84 | 0.77 | 0.77 |
| fra_Latn | 0.92 | 0.84 | 0.94 | 0.88 | 0.87 | 0.37 | 0.41 | 0.84 | 0.70 | 0.75 |
| por_Latn | 0.92 | 0.84 | 0.94 | 0.88 | 0.87 | 0.41 | 0.41 | 0.84 | 0.63 | 0.75 |
| deu_Latn | 0.92 | 0.84 | 0.94 | 0.88 | 0.87 | 0.35 | 0.42 | 0.84 | 0.65 | 0.74 |
| spa_Latn | 0.92 | 0.83 | 0.95 | 0.88 | 0.87 | 0.37 | 0.42 | 0.84 | 0.56 | 0.74 |
| ita_Latn | 0.92 | 0.83 | 0.92 | 0.88 | 0.87 | 0.35 | 0.42 | 0.84 | 0.56 | 0.73 |
| cat_Latn | 0.92 | 0.82 | 0.94 | 0.88 | 0.87 | 0.39 | 0.42 | 0.84 | 0.50 | 0.73 |
| nld_Latn | 0.92 | 0.82 | 0.95 | 0.88 | 0.87 | 0.34 | 0.42 | 0.84 | 0.52 | 0.73 |
| rus_Cyrl | 0.91 | 0.82 | 0.94 | 0.88 | 0.87 | 0.34 | 0.41 | 0.83 | 0.51 | 0.72 |
| zho_Hans | 0.91 | 0.80 | 0.94 | 0.88 | 0.87 | 0.32 | 0.34 | 0.81 | 0.62 | 0.72 |
| glg_Latn | 0.92 | 0.83 | 0.91 | 0.88 | 0.87 | 0.31 | 0.41 | 0.82 | 0.52 | 0.72 |
| swe_Latn | 0.92 | 0.83 | 0.95 | 0.88 | 0.87 | 0.38 | 0.42 | 0.84 | 0.37 | 0.72 |
| dan_Latn | 0.92 | 0.83 | 0.94 | 0.88 | 0.87 | 0.31 | 0.41 | 0.82 | 0.44 | 0.71 |
| ces_Latn | 0.92 | 0.82 | 0.95 | 0.88 | 0.87 | 0.26 | 0.42 | 0.84 | 0.43 | 0.71 |
| ron_Latn | 0.92 | 0.82 | 0.94 | 0.88 | 0.87 | 0.23 | 0.41 | 0.83 | 0.48 | 0.71 |
| nob_Latn | 0.91 | 0.82 | 0.95 | 0.88 | 0.87 | 0.34 | 0.39 | 0.81 | 0.39 | 0.71 |
| zho_Hant | 0.91 | 0.81 | 0.94 | 0.88 | 0.87 | 0.31 | 0.32 | 0.79 | 0.52 | 0.71 |
| pol_Latn | 0.92 | 0.81 | 0.95 | 0.88 | 0.87 | 0.22 | 0.42 | 0.84 | 0.38 | 0.70 |

Continued on next page

| | Gemma 2 9B | Gemma 1 7B | Llama 3.1 70B | Llama 3.1 8B | Llama 3 8B | Llama 2 7B | Llama 1 7B | Mistral 7B | OLMo 7B | AVG |
|---|---|---|---|---|---|---|---|---|---|---|
| ast_Latn | 0.90 | 0.80 | 0.91 | 0.88 | 0.86 | 0.21 | 0.40 | 0.77 | 0.49 | 0.69 |
| ind_Latn | 0.92 | 0.83 | 0.93 | 0.87 | 0.87 | 0.22 | 0.30 | 0.82 | 0.42 | 0.69 |
| oci_Latn | 0.89 | 0.75 | 0.95 | 0.88 | 0.87 | 0.22 | 0.39 | 0.81 | 0.40 | 0.68 |
| bos_Latn | 0.91 | 0.81 | 0.95 | 0.88 | 0.87 | 0.19 | 0.41 | 0.84 | 0.25 | 0.68 |
| nno_Latn | 0.92 | 0.82 | 0.92 | 0.84 | 0.84 | 0.26 | 0.36 | 0.78 | 0.38 | 0.68 |
| ukr_Cyrl | 0.92 | 0.81 | 0.95 | 0.88 | 0.87 | 0.22 | 0.42 | 0.84 | 0.15 | 0.67 |
| zsm_Latn | 0.92 | 0.83 | 0.93 | 0.88 | 0.87 | 0.17 | 0.25 | 0.81 | 0.36 | 0.67 |
| hrv_Latn | 0.91 | 0.81 | 0.90 | 0.86 | 0.86 | 0.18 | 0.41 | 0.83 | 0.23 | 0.67 |
| slv_Latn | 0.91 | 0.79 | 0.93 | 0.86 | 0.86 | 0.20 | 0.40 | 0.84 | 0.19 | 0.66 |
| afr_Latn | 0.91 | 0.81 | 0.93 | 0.87 | 0.87 | 0.20 | 0.37 | 0.79 | 0.21 | 0.66 |
| slk_Latn | 0.91 | 0.80 | 0.93 | 0.86 | 0.85 | 0.12 | 0.38 | 0.82 | 0.25 | 0.66 |
| bul_Cyrl | 0.91 | 0.80 | 0.90 | 0.86 | 0.86 | 0.12 | 0.42 | 0.84 | 0.14 | 0.65 |
| jpn_Jpan | 0.90 | 0.80 | 0.93 | 0.83 | 0.82 | 0.29 | 0.25 | 0.76 | 0.24 | 0.65 |
| hun_Latn | 0.91 | 0.78 | 0.92 | 0.84 | 0.83 | 0.13 | 0.39 | 0.81 | 0.18 | 0.64 |
| vec_Latn | 0.87 | 0.74 | 0.93 | 0.84 | 0.83 | 0.16 | 0.35 | 0.76 | 0.28 | 0.64 |
| srp_Cyrl | 0.91 | 0.79 | 0.90 | 0.86 | 0.86 | 0.10 | 0.42 | 0.84 | 0.06 | 0.64 |
| tgl_Latn | 0.91 | 0.74 | 0.94 | 0.82 | 0.82 | 0.16 | 0.20 | 0.77 | 0.36 | 0.64 |
| fin_Latn | 0.91 | 0.79 | 0.90 | 0.85 | 0.85 | 0.14 | 0.21 | 0.74 | 0.33 | 0.64 |
| mkd_Cyrl | 0.90 | 0.77 | 0.94 | 0.87 | 0.86 | 0.07 | 0.38 | 0.80 | 0.11 | 0.63 |
| vie_Latn | 0.91 | 0.81 | 0.95 | 0.88 | 0.87 | 0.22 | 0.08 | 0.79 | 0.16 | 0.63 |
| epo_Latn | 0.87 | 0.76 | 0.95 | 0.86 | 0.85 | 0.14 | 0.26 | 0.67 | 0.11 | 0.61 |
| kor_Hang | 0.88 | 0.74 | 0.91 | 0.84 | 0.83 | 0.22 | 0.15 | 0.71 | 0.15 | 0.60 |
| arb_Arab | 0.91 | 0.80 | 0.94 | 0.86 | 0.85 | 0.05 | 0.17 | 0.70 | 0.10 | 0.60 |
| ars_Arab | 0.91 | 0.80 | 0.93 | 0.86 | 0.85 | 0.04 | 0.17 | 0.69 | 0.08 | 0.59 |
| lim_Latn | 0.76 | 0.65 | 0.89 | 0.83 | 0.83 | 0.21 | 0.25 | 0.59 | 0.21 | 0.58 |
| acq_Arab | 0.91 | 0.78 | 0.92 | 0.83 | 0.82 | 0.04 | 0.13 | 0.67 | 0.09 | 0.58 |
| acm_Arab | 0.90 | 0.76 | 0.90 | 0.86 | 0.82 | 0.04 | 0.14 | 0.67 | 0.09 | 0.57 |
| fur_Latn | 0.73 | 0.60 | 0.91 | 0.81 | 0.77 | 0.16 | 0.27 | 0.59 | 0.26 | 0.57 |
| pes_Arab | 0.91 | 0.79 | 0.88 | 0.85 | 0.85 | 0.05 | 0.08 | 0.59 | 0.07 | 0.56 |
| arz_Arab | 0.88 | 0.74 | 0.90 | 0.84 | 0.83 | 0.03 | 0.10 | 0.63 | 0.09 | 0.56 |
| ajp_Arab | 0.88 | 0.76 | 0.86 | 0.85 | 0.83 | 0.03 | 0.12 | 0.60 | 0.09 | 0.56 |
| lit_Latn | 0.90 | 0.76 | 0.92 | 0.78 | 0.80 | 0.10 | 0.10 | 0.56 | 0.11 | 0.56 |
| apc_Arab | 0.89 | 0.76 | 0.86 | 0.82 | 0.83 | 0.03 | 0.11 | 0.64 | 0.09 | 0.56 |
| ell_Grek | 0.90 | 0.78 | 0.87 | 0.87 | 0.86 | 0.02 | 0.09 | 0.58 | 0.05 | 0.56 |
| tur_Latn | 0.89 | 0.78 | 0.90 | 0.82 | 0.81 | 0.04 | 0.09 | 0.61 | 0.04 | 0.55 |
| est_Latn | 0.90 | 0.77 | 0.90 | 0.82 | 0.83 | 0.12 | 0.09 | 0.45 | 0.10 | 0.55 |
| pap_Latn | 0.79 | 0.60 | 0.89 | 0.75 | 0.73 | 0.18 | 0.22 | 0.56 | 0.23 | 0.55 |
| lmo_Latn | 0.73 | 0.56 | 0.87 | 0.75 | 0.74 | 0.17 | 0.26 | 0.60 | 0.26 | 0.55 |
| szl_Latn | 0.77 | 0.59 | 0.87 | 0.73 | 0.74 | 0.11 | 0.26 | 0.64 | 0.21 | 0.55 |
| prs_Arab | 0.90 | 0.78 | 0.92 | 0.84 | 0.84 | 0.01 | 0.06 | 0.46 | 0.08 | 0.54 |
| scn_Latn | 0.77 | 0.59 | 0.88 | 0.79 | 0.77 | 0.15 | 0.22 | 0.57 | 0.15 | 0.54 |
| heb_Hebr | 0.91 | 0.81 | 0.89 | 0.83 | 0.83 | 0.02 | 0.05 | 0.47 | 0.06 | 0.54 |
| lvs_Latn | 0.90 | 0.75 | 0.90 | 0.81 | 0.79 | 0.05 | 0.05 | 0.55 | 0.08 | 0.54 |
| als_Latn | 0.87 | 0.67 | 0.93 | 0.79 | 0.80 | 0.09 | 0.08 | 0.53 | 0.10 | 0.54 |
| lij_Latn | 0.74 | 0.58 | 0.88 | 0.72 | 0.70 | 0.16 | 0.25 | 0.53 | 0.30 | 0.54 |
| ceb_Latn | 0.83 | 0.59 | 0.89 | 0.73 | 0.72 | 0.16 | 0.15 | 0.49 | 0.24 | 0.53 |
| srd_Latn | 0.73 | 0.59 | 0.86 | 0.75 | 0.72 | 0.16 | 0.23 | 0.55 | 0.21 | 0.53 |
| hin_Deva | 0.90 | 0.74 | 0.91 | 0.80 | 0.79 | 0.03 | 0.05 | 0.44 | 0.06 | 0.53 |
| ltz_Latn | 0.79 | 0.59 | 0.84 | 0.75 | 0.74 | 0.15 | 0.18 | 0.44 | 0.18 | 0.52 |
| tha_Thai | 0.90 | 0.76 | 0.87 | 0.83 | 0.83 | 0.02 | 0.02 | 0.32 | 0.10 | 0.52 |
| aeb_Arab | 0.82 | 0.67 | 0.86 | 0.78 | 0.75 | 0.04 | 0.10 | 0.55 | 0.08 | 0.52 |
| bel_Cyrl | 0.88 | 0.65 | 0.88 | 0.79 | 0.79 | 0.02 | 0.09 | 0.50 | 0.02 | 0.51 |
| isl_Latn | 0.83 | 0.62 | 0.88 | 0.77 | 0.78 | 0.09 | 0.10 | 0.48 | 0.06 | 0.51 |
| swh_Latn | 0.90 | 0.74 | 0.86 | 0.73 | 0.80 | 0.11 | 0.09 | 0.27 | 0.08 | 0.51 |
| mlt_Latn | 0.88 | 0.63 | 0.87 | 0.74 | 0.74 | 0.12 | 0.11 | 0.38 | 0.12 | 0.51 |
| war_Latn | 0.76 | 0.55 | 0.88 | 0.65 | 0.61 | 0.26 | 0.20 | 0.35 | 0.20 | 0.49 |
| cym_Latn | 0.87 | 0.59 | 0.88 | 0.75 | 0.76 | 0.11 | 0.10 | 0.28 | 0.08 | 0.49 |
| fao_Latn | 0.71 | 0.53 | 0.86 | 0.71 | 0.69 | 0.12 | 0.13 | 0.53 | 0.08 | 0.48 |
| urd_Arab | 0.83 | 0.66 | 0.88 | 0.76 | 0.73 | 0.02 | 0.02 | 0.31 | 0.03 | 0.47 |
| jav_Latn | 0.75 | 0.54 | 0.84 | 0.69 | 0.67 | 0.16 | 0.12 | 0.29 | 0.16 | 0.47 |

| | Gemma 2 9B | Gemma 1 7B | Llama 3.1 70B | Llama 3.1 8B | Llama 3 8B | Llama 2 7B | Llama 1 7B | Mistral 7B | OLMo 7B | AVG |
|---|---|---|---|---|---|---|---|---|---|---|
| eus_Latn | 0.82 | 0.66 | 0.84 | 0.74 | 0.71 | 0.10 | 0.08 | 0.18 | 0.05 | 0.47 |
| sun_Latn | 0.69 | 0.48 | 0.86 | 0.68 | 0.64 | 0.14 | 0.12 | 0.34 | 0.23 | 0.47 |
| kea_Latn | 0.64 | 0.51 | 0.78 | 0.60 | 0.64 | 0.16 | 0.19 | 0.45 | 0.18 | 0.46 |
| ary_Arab | 0.71 | 0.60 | 0.80 | 0.68 | 0.68 | 0.03 | 0.09 | 0.44 | 0.10 | 0.46 |
| hat_Latn | 0.74 | 0.47 | 0.86 | 0.65 | 0.61 | 0.15 | 0.12 | 0.36 | 0.09 | 0.45 |
| mag_Deva | 0.75 | 0.52 | 0.88 | 0.75 | 0.70 | 0.02 | 0.05 | 0.29 | 0.07 | 0.45 |
| min_Latn | 0.56 | 0.45 | 0.81 | 0.68 | 0.68 | 0.16 | 0.12 | 0.34 | 0.17 | 0.44 |
| ban_Latn | 0.48 | 0.32 | 0.82 | 0.62 | 0.63 | 0.17 | 0.15 | 0.42 | 0.30 | 0.43 |
| bjn_Latn | 0.60 | 0.51 | 0.78 | 0.62 | 0.60 | 0.14 | 0.12 | 0.31 | 0.21 | 0.43 |
| azj_Latn | 0.75 | 0.53 | 0.86 | 0.68 | 0.66 | 0.01 | 0.03 | 0.29 | 0.02 | 0.43 |
| npi_Deva | 0.81 | 0.59 | 0.82 | 0.62 | 0.64 | 0.02 | 0.02 | 0.20 | 0.05 | 0.42 |
| mar_Deva | 0.82 | 0.58 | 0.87 | 0.68 | 0.63 | 0.02 | 0.01 | 0.12 | 0.03 | 0.42 |
| awa_Deva | 0.73 | 0.53 | 0.83 | 0.65 | 0.65 | 0.02 | 0.05 | 0.21 | 0.08 | 0.42 |
| ben_Beng | 0.82 | 0.58 | 0.82 | 0.60 | 0.60 | 0.01 | 0.01 | 0.23 | 0.05 | 0.41 |
| uzn_Latn | 0.70 | 0.47 | 0.84 | 0.60 | 0.62 | 0.04 | 0.05 | 0.23 | 0.05 | 0.40 |
| bho_Deva | 0.51 | 0.46 | 0.89 | 0.68 | 0.65 | 0.02 | 0.03 | 0.26 | 0.08 | 0.40 |
| gle_Latn | 0.68 | 0.31 | 0.82 | 0.64 | 0.64 | 0.09 | 0.08 | 0.22 | 0.09 | 0.40 |
| hye_Armn | 0.85 | 0.59 | 0.79 | 0.60 | 0.58 | 0.01 | 0.02 | 0.10 | 0.01 | 0.40 |
| hne_Deva | 0.67 | 0.44 | 0.80 | 0.65 | 0.63 | 0.01 | 0.02 | 0.24 | 0.08 | 0.40 |
| kaz_Cyrl | 0.62 | 0.47 | 0.87 | 0.61 | 0.60 | 0.02 | 0.06 | 0.27 | 0.03 | 0.39 |
| tpi_Latn | 0.69 | 0.38 | 0.69 | 0.46 | 0.45 | 0.19 | 0.15 | 0.33 | 0.15 | 0.39 |
| hau_Latn | 0.68 | 0.41 | 0.77 | 0.58 | 0.54 | 0.12 | 0.11 | 0.17 | 0.06 | 0.38 |
| mai_Deva | 0.61 | 0.43 | 0.86 | 0.61 | 0.59 | 0.02 | 0.03 | 0.16 | 0.08 | 0.38 |
| crh_Latn | 0.58 | 0.42 | 0.77 | 0.56 | 0.51 | 0.03 | 0.07 | 0.35 | 0.05 | 0.37 |
| ilo_Latn | 0.61 | 0.32 | 0.74 | 0.47 | 0.46 | 0.17 | 0.15 | 0.24 | 0.13 | 0.37 |
| tat_Cyrl | 0.56 | 0.38 | 0.83 | 0.58 | 0.55 | 0.03 | 0.04 | 0.21 | 0.02 | 0.36 |
| kat_Geor | 0.73 | 0.45 | 0.72 | 0.53 | 0.50 | 0.01 | 0.04 | 0.18 | 0.01 | 0.35 |
| ydd_Hebr | 0.74 | 0.45 | 0.78 | 0.48 | 0.47 | 0.03 | 0.02 | 0.05 | 0.02 | 0.34 |
| kir_Cyrl | 0.53 | 0.33 | 0.81 | 0.58 | 0.57 | 0.01 | 0.04 | 0.13 | 0.02 | 0.34 |
| pag_Latn | 0.33 | 0.23 | 0.63 | 0.35 | 0.39 | 0.25 | 0.21 | 0.36 | 0.24 | 0.33 |
| pan_Guru | 0.78 | 0.50 | 0.75 | 0.47 | 0.40 | 0.01 | 0.00 | 0.03 | 0.05 | 0.33 |
| bak_Cyrl | 0.56 | 0.36 | 0.82 | 0.51 | 0.51 | 0.01 | 0.04 | 0.13 | 0.02 | 0.33 |
| guj_Gujr | 0.79 | 0.52 | 0.67 | 0.42 | 0.39 | 0.02 | 0.01 | 0.04 | 0.07 | 0.33 |
| tam_Taml | 0.78 | 0.54 | 0.72 | 0.38 | 0.38 | 0.02 | 0.01 | 0.08 | 0.03 | 0.33 |
| pbt_Arab | 0.50 | 0.23 | 0.82 | 0.57 | 0.57 | 0.02 | 0.03 | 0.10 | 0.07 | 0.32 |
| tgk_Cyrl | 0.62 | 0.27 | 0.78 | 0.51 | 0.52 | 0.02 | 0.03 | 0.10 | 0.05 | 0.32 |
| tel_Telu | 0.77 | 0.52 | 0.67 | 0.38 | 0.43 | 0.01 | 0.01 | 0.05 | 0.04 | 0.32 |
| snd_Arab | 0.59 | 0.30 | 0.78 | 0.53 | 0.50 | 0.02 | 0.01 | 0.06 | 0.04 | 0.31 |
| kan_Knda | 0.74 | 0.47 | 0.66 | 0.42 | 0.41 | 0.01 | 0.01 | 0.06 | 0.05 | 0.31 |
| mal_Mlym | 0.76 | 0.50 | 0.68 | 0.32 | 0.30 | 0.01 | 0.01 | 0.03 | 0.03 | 0.29 |
| ckb_Arab | 0.51 | 0.20 | 0.78 | 0.54 | 0.50 | 0.01 | 0.01 | 0.05 | 0.02 | 0.29 |
| gla_Latn | 0.46 | 0.14 | 0.71 | 0.46 | 0.45 | 0.08 | 0.07 | 0.13 | 0.07 | 0.29 |
| asm_Beng | 0.63 | 0.35 | 0.70 | 0.39 | 0.36 | 0.00 | 0.01 | 0.08 | 0.04 | 0.29 |
| tuk_Latn | 0.49 | 0.31 | 0.63 | 0.43 | 0.41 | 0.05 | 0.04 | 0.14 | 0.02 | 0.28 |
| san_Deva | 0.48 | 0.26 | 0.71 | 0.46 | 0.45 | 0.00 | 0.01 | 0.12 | 0.02 | 0.28 |
| kmr_Latn | 0.38 | 0.15 | 0.69 | 0.48 | 0.50 | 0.05 | 0.06 | 0.13 | 0.05 | 0.28 |
| lus_Latn | 0.53 | 0.09 | 0.56 | 0.34 | 0.33 | 0.14 | 0.10 | 0.24 | 0.11 | 0.27 |
| khk_Cyrl | 0.44 | 0.18 | 0.73 | 0.42 | 0.43 | 0.01 | 0.02 | 0.08 | 0.03 | 0.26 |
| ltg_Latn | 0.31 | 0.23 | 0.61 | 0.38 | 0.35 | 0.08 | 0.06 | 0.22 | 0.06 | 0.26 |
| azb_Arab | 0.37 | 0.28 | 0.60 | 0.44 | 0.44 | 0.00 | 0.01 | 0.11 | 0.02 | 0.25 |
| plt_Latn | 0.52 | 0.17 | 0.59 | 0.25 | 0.25 | 0.14 | 0.12 | 0.18 | 0.05 | 0.25 |
| ibo_Latn | 0.35 | 0.15 | 0.64 | 0.38 | 0.37 | 0.09 | 0.08 | 0.12 | 0.07 | 0.25 |
| mri_Latn | 0.35 | 0.11 | 0.60 | 0.38 | 0.35 | 0.12 | 0.10 | 0.18 | 0.07 | 0.25 |
| som_Latn | 0.42 | 0.14 | 0.60 | 0.24 | 0.24 | 0.11 | 0.08 | 0.18 | 0.09 | 0.23 |
| ace_Latn | 0.22 | 0.13 | 0.49 | 0.32 | 0.31 | 0.14 | 0.10 | 0.21 | 0.15 | 0.23 |
| xho_Latn | 0.49 | 0.19 | 0.47 | 0.20 | 0.20 | 0.12 | 0.10 | 0.13 | 0.05 | 0.22 |
| nso_Latn | 0.27 | 0.11 | 0.48 | 0.26 | 0.23 | 0.17 | 0.13 | 0.19 | 0.07 | 0.21 |
| sot_Latn | 0.34 | 0.12 | 0.53 | 0.22 | 0.20 | 0.14 | 0.10 | 0.18 | 0.05 | 0.21 |
| zul_Latn | 0.55 | 0.19 | 0.44 | 0.19 | 0.17 | 0.11 | 0.06 | 0.10 | 0.05 | 0.21 |
| kin_Latn | 0.37 | 0.10 | 0.53 | 0.20 | 0.23 | 0.11 | 0.09 | 0.15 | 0.06 | 0.21 |

| | Gemma 2 9B | Gemma 1 7B | Llama 3.1 70B | Llama 3.1 8B | Llama 3 8B | Llama 2 7B | Llama 1 7B | Mistral 7B | OLMo 7B | AVG |
|---|---|---|---|---|---|---|---|---|---|---|
| sin_Sinh | 0.56 | 0.27 | 0.49 | 0.26 | 0.18 | 0.01 | 0.01 | 0.03 | 0.02 | 0.20 |
| smo_Latn | 0.28 | 0.09 | 0.66 | 0.20 | 0.20 | 0.10 | 0.08 | 0.13 | 0.06 | 0.20 |
| nya_Latn | 0.36 | 0.13 | 0.41 | 0.19 | 0.19 | 0.13 | 0.10 | 0.17 | 0.06 | 0.19 |
| twi_Latn | 0.23 | 0.08 | 0.46 | 0.22 | 0.22 | 0.14 | 0.13 | 0.19 | 0.07 | 0.19 |
| sna_Latn | 0.41 | 0.17 | 0.40 | 0.19 | 0.20 | 0.10 | 0.08 | 0.12 | 0.07 | 0.19 |
| uig_Arab | 0.21 | 0.09 | 0.71 | 0.29 | 0.29 | 0.00 | 0.01 | 0.03 | 0.02 | 0.18 |
| bug_Latn | 0.14 | 0.12 | 0.35 | 0.22 | 0.22 | 0.14 | 0.11 | 0.20 | 0.12 | 0.18 |
| luo_Latn | 0.07 | 0.07 | 0.40 | 0.25 | 0.24 | 0.15 | 0.12 | 0.21 | 0.09 | 0.18 |
| tsn_Latn | 0.24 | 0.10 | 0.42 | 0.18 | 0.18 | 0.12 | 0.11 | 0.20 | 0.06 | 0.18 |
| arb_Latn | 0.29 | 0.07 | 0.46 | 0.24 | 0.20 | 0.05 | 0.05 | 0.17 | 0.08 | 0.18 |
| khm_Khmr | 0.34 | 0.15 | 0.59 | 0.15 | 0.16 | 0.01 | 0.02 | 0.09 | 0.06 | 0.17 |
| lua_Latn | 0.09 | 0.08 | 0.33 | 0.20 | 0.21 | 0.14 | 0.13 | 0.24 | 0.12 | 0.17 |
| lug_Latn | 0.17 | 0.07 | 0.41 | 0.18 | 0.19 | 0.14 | 0.09 | 0.19 | 0.06 | 0.17 |
| grn_Latn | 0.17 | 0.09 | 0.44 | 0.16 | 0.17 | 0.12 | 0.09 | 0.13 | 0.10 | 0.16 |
| ssw_Latn | 0.27 | 0.10 | 0.37 | 0.17 | 0.17 | 0.11 | 0.08 | 0.14 | 0.05 | 0.16 |
| lin_Latn | 0.11 | 0.08 | 0.43 | 0.16 | 0.18 | 0.12 | 0.11 | 0.16 | 0.08 | 0.16 |
| ory_Orya | 0.28 | 0.03 | 0.66 | 0.18 | 0.19 | 0.01 | 0.01 | 0.03 | 0.03 | 0.16 |
| fij_Latn | 0.13 | 0.06 | 0.38 | 0.18 | 0.16 | 0.14 | 0.11 | 0.15 | 0.08 | 0.15 |
| fuv_Latn | 0.07 | 0.08 | 0.30 | 0.20 | 0.20 | 0.13 | 0.10 | 0.18 | 0.10 | 0.15 |
| kas_Arab | 0.16 | 0.10 | 0.50 | 0.20 | 0.21 | 0.02 | 0.02 | 0.10 | 0.05 | 0.15 |
| quy_Latn | 0.10 | 0.06 | 0.42 | 0.21 | 0.22 | 0.10 | 0.06 | 0.13 | 0.05 | 0.15 |
| aka_Latn | 0.17 | 0.06 | 0.37 | 0.14 | 0.17 | 0.11 | 0.08 | 0.12 | 0.10 | 0.15 |
| mya_Mymr | 0.36 | 0.13 | 0.46 | 0.14 | 0.16 | 0.00 | 0.00 | 0.02 | 0.02 | 0.15 |
| run_Latn | 0.25 | 0.07 | 0.37 | 0.16 | 0.17 | 0.08 | 0.06 | 0.10 | 0.04 | 0.14 |
| bem_Latn | 0.14 | 0.08 | 0.29 | 0.16 | 0.16 | 0.13 | 0.11 | 0.15 | 0.06 | 0.14 |
| kas_Deva | 0.14 | 0.09 | 0.37 | 0.20 | 0.21 | 0.02 | 0.03 | 0.11 | 0.08 | 0.14 |
| wol_Latn | 0.09 | 0.07 | 0.30 | 0.18 | 0.16 | 0.12 | 0.10 | 0.17 | 0.07 | 0.14 |
| kam_Latn | 0.09 | 0.08 | 0.26 | 0.18 | 0.16 | 0.13 | 0.10 | 0.15 | 0.08 | 0.14 |
| tso_Latn | 0.14 | 0.06 | 0.35 | 0.14 | 0.13 | 0.11 | 0.08 | 0.13 | 0.06 | 0.13 |
| kon_Latn | 0.07 | 0.08 | 0.27 | 0.15 | 0.17 | 0.09 | 0.07 | 0.13 | 0.09 | 0.13 |
| tum_Latn | 0.15 | 0.07 | 0.32 | 0.11 | 0.13 | 0.09 | 0.08 | 0.11 | 0.05 | 0.13 |
| kik_Latn | 0.07 | 0.04 | 0.32 | 0.12 | 0.13 | 0.10 | 0.09 | 0.13 | 0.12 | 0.12 |
| taq_Latn | 0.06 | 0.06 | 0.28 | 0.14 | 0.12 | 0.11 | 0.08 | 0.14 | 0.05 | 0.12 |
| mos_Latn | 0.04 | 0.04 | 0.25 | 0.16 | 0.14 | 0.11 | 0.09 | 0.15 | 0.07 | 0.12 |
| yor_Latn | 0.13 | 0.04 | 0.30 | 0.14 | 0.14 | 0.08 | 0.06 | 0.10 | 0.04 | 0.11 |
| amh_Ethi | 0.48 | 0.16 | 0.24 | 0.04 | 0.03 | 0.01 | 0.02 | 0.03 | 0.02 | 0.11 |
| sag_Latn | 0.05 | 0.07 | 0.22 | 0.17 | 0.17 | 0.07 | 0.07 | 0.09 | 0.06 | 0.11 |
| cjk_Latn | 0.06 | 0.06 | 0.21 | 0.13 | 0.12 | 0.10 | 0.08 | 0.13 | 0.07 | 0.11 |
| umb_Latn | 0.05 | 0.05 | 0.20 | 0.15 | 0.14 | 0.10 | 0.08 | 0.11 | 0.05 | 0.10 |
| dyu_Latn | 0.04 | 0.04 | 0.22 | 0.13 | 0.12 | 0.06 | 0.07 | 0.10 | 0.08 | 0.10 |
| kac_Latn | 0.02 | 0.03 | 0.22 | 0.12 | 0.14 | 0.06 | 0.06 | 0.12 | 0.08 | 0.09 |
| kmb_Latn | 0.05 | 0.06 | 0.20 | 0.11 | 0.10 | 0.10 | 0.07 | 0.09 | 0.05 | 0.09 |
| bam_Latn | 0.05 | 0.05 | 0.18 | 0.11 | 0.09 | 0.08 | 0.08 | 0.12 | 0.04 | 0.09 |
| ayr_Latn | 0.04 | 0.04 | 0.20 | 0.11 | 0.10 | 0.06 | 0.05 | 0.10 | 0.06 | 0.08 |
| lao_Laoo | 0.17 | 0.04 | 0.22 | 0.07 | 0.09 | 0.02 | 0.02 | 0.04 | 0.09 | 0.08 |
| dik_Latn | 0.05 | 0.06 | 0.18 | 0.06 | 0.07 | 0.08 | 0.07 | 0.10 | 0.05 | 0.08 |
| ewe_Latn | 0.04 | 0.03 | 0.18 | 0.08 | 0.08 | 0.09 | 0.07 | 0.08 | 0.04 | 0.08 |
| knc_Latn | 0.05 | 0.06 | 0.15 | 0.08 | 0.08 | 0.07 | 0.06 | 0.08 | 0.05 | 0.08 |
| kab_Latn | 0.04 | 0.02 | 0.17 | 0.09 | 0.08 | 0.06 | 0.06 | 0.11 | 0.04 | 0.07 |
| sat_Olck | 0.19 | 0.02 | 0.32 | 0.05 | 0.05 | 0.00 | 0.00 | 0.01 | 0.01 | 0.07 |
| gaz_Latn | 0.05 | 0.03 | 0.20 | 0.06 | 0.06 | 0.05 | 0.04 | 0.08 | 0.03 | 0.07 |
| bod_Tibt | 0.07 | 0.01 | 0.22 | 0.08 | 0.08 | 0.01 | 0.01 | 0.02 | 0.02 | 0.06 |
| fon_Latn | 0.03 | 0.02 | 0.14 | 0.06 | 0.06 | 0.03 | 0.04 | 0.05 | 0.07 | 0.06 |
| shn_Mymr | 0.02 | 0.01 | 0.21 | 0.06 | 0.06 | 0.01 | 0.02 | 0.03 | 0.07 | 0.05 |
| kbp_Latn | 0.03 | 0.02 | 0.14 | 0.05 | 0.04 | 0.03 | 0.02 | 0.08 | 0.05 | 0.05 |
| mni_Beng | 0.03 | 0.02 | 0.12 | 0.05 | 0.06 | 0.01 | 0.02 | 0.08 | 0.02 | 0.05 |
| ace_Arab | 0.03 | 0.02 | 0.15 | 0.07 | 0.07 | 0.00 | 0.00 | 0.03 | 0.01 | 0.04 |
| knc_Arab | 0.01 | 0.01 | 0.13 | 0.04 | 0.04 | 0.02 | 0.02 | 0.05 | 0.05 | 0.04 |
| bjn_Arab | 0.03 | 0.02 | 0.11 | 0.05 | 0.08 | 0.01 | 0.01 | 0.05 | 0.01 | 0.04 |
| nus_Latn | 0.02 | 0.02 | 0.07 | 0.04 | 0.03 | 0.03 | 0.02 | 0.04 | 0.05 | 0.03 |

|          | Gemma 2 9B | Gemma 1 7B | Llama 3.1 70B | Llama 3.1 8B | Llama 3 8B | Llama 2 7B | Llama 1 7B | Mistral 7B | OLMo 7B | AVG |
|----------|------------|------------|---------------|--------------|------------|------------|------------|------------|---------|-----|
| **min_Arab** | 0.02 | 0.01 | 0.13 | 0.05 | 0.04 | 0.01 | 0.00 | 0.03 | 0.01 | 0.03 |
| **tir_Ethi** | 0.10 | 0.02 | 0.05 | 0.02 | 0.02 | 0.01 | 0.01 | 0.02 | 0.02 | 0.03 |
| **dzo_Tibt** | 0.01 | 0.00 | 0.08 | 0.03 | 0.03 | 0.00 | 0.00 | 0.01 | 0.01 | 0.02 |
| **taq_Tfng** | 0.00 | 0.00 | 0.04 | 0.01 | 0.01 | 0.00 | 0.00 | 0.01 | 0.03 | 0.01 |
| **zgh_Tfng** | 0.00 | 0.00 | 0.02 | 0.01 | 0.01 | 0.00 | 0.00 | 0.01 | 0.01 | 0.01 |

Table 5: Adjusted performance of MEXA using max pooling with the English performance of models on the Belebele benchmark.

