# OpenReview forum: "Mexa: Multilingual Evaluation of English-Centric LLMs via  Cross-Lingual Alignment"
_ICLR.cc/2025/Conference — ICLR 2025 Conference Withdrawn Submission_

### Official Review · Reviewer_oR5Z · 2024-10-30

**Soundness:** 2
**Presentation:** 3
**Contribution:** 3
**Rating:** 5
**Confidence:** 4

**Summary:**

This paper proposes Mexa, which is a leaderboard for estimating the cross-lingual abilities of English-based large language models. Mexa works by using a large corpus of parallel sentences (e.g., FLORES or Bible corpora) and calculating the semantic similarity score between the given model's embeddings of the English and non-English parallel sentences. These are considered aligned if the similarity between the parallel sentences is greater than between any two non-parallel sentences. This score is aggregated over the whole corpus, and it is found to be a strong predictor of performance on downstream non-English tasks (modulo the corresponding English performance on the task). This is a useful benchmark because parallel sentences and English benchmarks are widely available, but multilingual benchmarks are less readily available.

**Strengths:**

1. The proposed Mexa approach is interesting and novel. I could see this approach inspiring future work in analysis of multilingual LLM performance.

2. It is impressively accurate at predicting the performance of several decoder-only English-centric LLMs on 3 different multilingual datasets

**Weaknesses:**

1. Lack of variety in LLMs considered. All LLMs evaluated are quite similar: they are all decoder-only models with very similar architectures. For a benchmark to be useful it needs to be reliable on arbitrary models including new ones that might be invented in the future, so it would be good to evaluate on a wider variety of architectures. Even something like Mixtral would be a good place to start. This is particularly important since some of the assumptions behind Mexa (e.g. in the paragraph at line 49-63, about English being the pivot language in the middle layers) seem based on analysis of Llama so may not hold as well when evaluating models with other architectures.

2. Benchmarks used. Mexa is quite good at predicting performance on the three multilingual benchmarks considered (table 2). However, it's not clear to me why only three benchmarks are considered; table 4 lists several more that are available. In particular, both m-MMLU and m-ARC rely on machine translation to generate the multilingual data, which means that they are more likely to align with the semantic similarity measures used, so I am not sure whether this generalizes to performance on realistic multilingual tasks. Also, there is a bit of tuning on these datasets (weighted average vs. last token, max pooling vs. mean pooling, FLORES vs. bible), and there is no held-out benchmark to truly evaluate generalization.

3. Inability to benchmark closed-source models. Unless I misunderstood, Mexa requires access to the weights to be able to evaluate a model. This makes it relatively impractical: practitioners may want to select a model by using Mexa to estimate performance on their desired task, but they will not be able to evaluate closed-source models.

**Questions:**

1. The idea of an "English-centric" LLM is referenced several times in the paper, but I do not see a concrete definition of "English-centric" anywhere in the paper. From my understanding, the intent of Mexa is to evaluate only English-centric LLMs, so it would be good to define what you mean by this so that practitioners know whether or not to use Mexa for a given LLM.

2. L252 "Although there are multilingual models [...] our focus here is on LLMs which are **state-of-the-art in English based tasks**": Why is the focus only on evaluating SOTA LLMs? The intention of Mexa is to be a public benchmark and leaderboard, so it will be used for a variety of LLMs, right? (including e.g. some that are really focused on one task so aren't SOTA across a variety of tasks). I am thinking it would be good to evaluate on a greater variety of LLMs for this purpose.

3. More generally, as a suggestion, I wonder if positioning this work as a benchmark was the best strategy. It was only evaluated on some relatively strict cases (open-weights/open-source models that are English-centric and decoder-only/Llama-esque and state-of-the-art on English-based tasks) which I think will hinder its usability as a benchmark (if evaluations do indeed need to be limited to such models). It may be better to position it as analysis?

---

> ### Author Response · Authors · 2024-11-25
>
> Thank you for your constructive review of our paper. We will refer to raised issues in the updated version of the paper, thanks!
>
> **Weaknesses:**
>
>
> > 1) Lack of variety in LLMs considered.
>
> Thanks for the reviewer’s suggestion. Our paper targets the widely used scenarios where the LLM is decoder-only and English-centric.
> Mixtral would serve as a starting point based on the reviewer’s suggestion. For Mixtral, it would not be significantly different in how we apply MEXA, as it applies to the hidden states at each layer for any given input. However, we will make sure to include this in the paper.
>
>
> > 2.1) Benchmarks used. Mexa is quite good at predicting performance on the three multilingual benchmarks considered (table 2). However, it's not clear to me why only three benchmarks are considered;
>
>
> We include all the possible tasks in Table 4 to explain why we chose these tasks. As you can see, for non-generative tasks, only these tasks support a high number of languages, including low-resource ones. Benchmarks with only 10 languages, mostly high-resource, would not support our claims, as MEXA would achieve high results for all of them. Belebele is the benchmark with the highest number of languages, while ARC and MMLU are the most important tasks on the community leaderboards for comparing models.
>
>
>
> > 2.2) Also, there is a bit of tuning on these datasets (weighted average vs. last token, max pooling vs. mean pooling, FLORES vs. bible), and there is no held-out benchmark to truly evaluate generalization.
>
>
> All of these settings are ablations to determine which works best in each scenario. To reduce confusion, we introduced one setting as the optimal choice. We explain the differences between these design choices in lines 361–388.
>
>
>
> > 3) Inability to benchmark closed-source models.
>
> True, this paper provides a method of evaluation for open science, and only model weights are needed. Although, developers of closed-source models could use MEXA under the hood and report their multilingual results to provide insight of their model's multilingual capabilities.
>
>
> **Questions:**
>
> > 1) The idea of an "English-centric" LLM is referenced several times in the paper, but I do not see a concrete definition of "English-centric" anywhere in the paper.
>
>
> Thanks for pointing that out! "English-centric" is not a new term used in this paper; it is also used in other papers (including https://arxiv.org/abs/2312.12683, https://arxiv.org/abs/2408.10811). Most of the models used in this paper did not disclose how much of their pre-training data is in English. We define English-centric decoder-only models as models where the latent language is English, which typically occurs when the majority of the data is in English. We will provide a clarified definition for this term in the paper.
>
>
> > 2) L252 "Although there are multilingual models [...] our focus here is on LLMs which are state-of-the-art in English based tasks": Why is the focus only on evaluating SOTA LLMs? The intention of Mexa is to be a public benchmark and leaderboard, so it will be used for a variety of LLMs, right?
>
> We tested on 9 different models. The English-centric models we chose span a range of performance levels, from weak to strong. For example, LLaMA 1 7B achieves a performance of 0.42 on English BeLEBELE, while LLaMA 3.1 reaches 0.95.
> Again, this selection of English-focused decoder models highlights those widely used in the community.
> We acknowledge that the text is ambiguous and will clarify this in the paper.
>
>
>
> > 3) More generally, as a suggestion, I wonder if positioning this work as a benchmark was the best strategy.
>
> We did not develop MEXA first and then search for a use case. Most of open science today focuses on English-centric decoder-only LLMs. However, their capabilities for many languages remain frustratingly undocumented. We developed a solution that provides a clearer vision and a rough estimate of the multilingual capabilities of these models.

---

> > ### Comment · Reviewer_oR5Z · 2024-11-25
> > **Response to author rebuttal**
> >
> > Thanks for your response! I will take a closer look, but since the discussion window is closing I wanted to ask two questions quickly:
> >
> > 1. In your response to weakness 2, I think you may have missed one concern I had: "In particular, both m-MMLU and m-ARC rely on machine translation to generate the multilingual data". Do you think using MT data is likely to be representative of the actual low-resource languages? Do you have any concerns about relying on MT data to benchmark low-resource languages?
> >
> > 2. You talk about "ablations to determine which works best in each scenario" and "design choices". What I find unclear in the paper is what this means for practitioners who want to use Mexa. Do you expect them to evaluate on both Flores and Bible? Will the choice of settings depend on their scenario? Etc.

---

> > > ### Author Response · Authors · 2024-11-25
> > >
> > > Thank you for the quick response.
> > >
> > > > 1. In your response to weakness 2, I think you may have missed one concern I had: "In particular, both m-MMLU and m-ARC rely on machine translation to generate the multilingual data". Do you think using MT data is likely to be representative of the actual low-resource languages? Do you have any concerns about relying on MT data to benchmark low-resource languages?
> > >
> > > Of course, there is always concern about MT-based data; it is definitely not perfect. However, we believe they are still good for evaluating trends in language performance. m-MMLU was also used in the Aya project (https://arxiv.org/abs/2402.07827) as a language understanding evaluation.
> > >
> > > > 2. You talk about "ablations to determine which works best in each scenario" and "design choices". What I find unclear in the paper is what this means for practitioners who want to use Mexa. Do you expect them to evaluate on both Flores and Bible? Will the choice of settings depend on their scenario? Etc.
> > >
> > > Depending on the available language resources, one may use only the Bible, FLORES, or both. We recommend using FLORES as it is a more conventional language benchmark . The rest of the settings remain unchanged. For the embeddings, always use weighted average embeddings. As for the pooling method, use both mean and max pooling, with mean pooling as the default. Max pooling reflects the upper bound of performance and is more correlated with easier tasks.

---

> > > > ### Author Response · Authors · 2024-11-26
> > > >
> > > > I would like to ask the reviewer if the response from the authors resolves the issues you raised. We would be happy to address any additional questions or suggestions you may have.

---

> > > > > ### Comment · Reviewer_oR5Z · 2024-12-02
> > > > > **Official comment**
> > > > >
> > > > > I want to thank the authors for their responses and discussion. Overall, my concerns about the usefulness of Mexa and the use of machine-translated data have not changed, so I will keep my score.

---

> > > > > > ### Author Response · Authors · 2024-12-02
> > > > > >
> > > > > > Thank you for your feedback. As said using translated MMLU and m-ARC is not ideal, and may bias some results for LR languages (or, more properly when the machine translation -typically with chapGPT- is poor). Yet these translated versions are representative of current state of automatic evaluation, as in https://huggingface.co/spaces/uonlp/open_multilingual_llm_leaderboard and https://github.com/nlp-uoregon/mlmm-evaluation.

---

### Official Review · Reviewer_bpzp · 2024-11-04

**Soundness:** 4
**Presentation:** 3
**Contribution:** 3
**Rating:** 6
**Confidence:** 5

**Summary:**

This paper proposes Mexa, a metric for quantifying cross-lingual alignment between language representations in LLMs. Most LLMs are English-centric and achieve multilingual performance in part by using English as a pivot language. This occurs in the middle layers of LLMs, where sentence representations are less language-specific, exhibiting cross-lingual alignment (similarity in vector representations). The degree of cross-lingual alignment of parallel sentences indicates to what extent English is utilised as a pivot language. Mexa quantifies this by comparing the cosine similarity of parallel sentences to non-parallel sentences, producing a metric that captures to what extent cross-lingual parallel sentences are representationally aligned. The authors present Mexa as a way to quantify cross-lingual transfer and estimate task performance based on how aligned languages are with English. Their results support the viability of this, with Mexa achieving comfortably higher correlation coefficients with downstream task performance than cosine similarity. The authors test different ways of computing sentence representations for Mexa, finding that different strategies are more effective for different types of downstream tasks.

**Strengths:**

(1) The main finding – that Mexa serves as a way to estimate downstream task performance based on cross-lingual alignment – is robustly supported by the results. Experiments were conducted across several LLMs, languages, and tasks, and Mexa exhibits very high correlation with task performance (both in terms of absolute correlation values and compared to cosine similarity.

(2) The authors have explored a number of design decisions for Mexa (which token representations to use to obtain sentence embeddings, how to pool embeddings) and present a full analysis of their findings. This provides practitioners with a clear guide of how to use and interpret Mexa in different contexts.

(3) The study conducted by the authors represent a valuable contribution to our understanding of multilingual LLMs. They build on previous findings and produce new insights into the mechanisms underlying cross-lingual transfers in LLMs.

**Weaknesses:**

(1) The authors seem to assume at the outset that cross-lingual representational similarity WILL entail better cross-lingual transfer. This assumption is the motivating factor behind Mexa. Their results do strongly suggest this, but they should be more cautious in stating this as a hypothesis (or assumption) at the start of the paper, or provide a survey of previous work proving the relation between alignment and downstream performance. It’s possible that alignment helps to some extent, but that too much cross-lingual overlap can impede performance e.g. for high-resource languages, where using English as a pivot might not be optimal. This might or might not be the case. My point is that the authors gloss over the assumption that more alignment implies better downstream performance for other languages. I see it more as a hypothesis that their findings strongly support.

(2) Mexa is not complex as a technical contribution. It is a simple way to encode relative cross-lingual similarity. While this in itself does not degrade its value as a contribution (its impact is clearly useful), it does not introduce impressive novelty from a technical standpoint.

**Questions:**

The writing in some parts of the paper could be made a bit clearer, specifically lines 139--147 and 197--202. Sentences like “This criterion imposes…”, “This suggests…”, “Another motivation…” are somewhat ambiguous in the first reading. It’s not clear if the authors are referring to their motivations for introducing a new concept or citing the motivations of previous research.

---

> ### Author Response · Authors · 2024-11-25
>
> Thank you for your very accurate review of our paper! All of the reviewer’s points are valid. We will revise the text in the introduction to reframe what we take as an assumption into a hypothesis strongly supported by the paper's findings. Additionally, we will clarify the lines mentioned that are somewhat ambiguous.

---

### Official Review · Reviewer_i2iS · 2024-11-04

**Soundness:** 3
**Presentation:** 3
**Contribution:** 3
**Rating:** 5
**Confidence:** 4

**Summary:**

This paper presents MEXA, a method to evaluate the multilingual capabilities of English-centric LLMs. Given the lack of multilingual benchmarks available, multilingual capabilities of English-centric LLMs may not be well evaluated across diverse languages.
MEXA relies on parallel sentences to calculate alignment scores between English and non-English languages to measure the transfer of language understanding from English to other languages. The authors ablate around optimal choices for layer and aggregation strategies in computing embeddings. The authors evaluate the MEXA across three multiple-choice style downstream tasks and observe that the MEXA scores demonstrates a strong correlation with downstream task performance.

**Strengths:**

The problem is very well motivated, particularly given the lack of multilingual benchmarks and such alignment based methods would be valuable to estimate multilingual performance in a resource-efficient manner. Such methods would be useful for determining model readiness for production deployment in specific languages.

The proposed method is straightforward and does not require any creation of additional sophisticated benchmarks or extra resources and is largely model-agnostic.

The paper is well-structured and easy to follow.

The authors perform exhaustive experiments across a wide range of languages and models, offering valuable, in-depth insights.

**Weaknesses:**

1. The MEXA formulation relies on relative scoring and does not account for the magnitude of cosine similarity. For example, in cases where scores are close to each other but also have low magnitudes, this minimal difference is inconsequential and does not necessarily imply alignment. Thus, the limitations of standard cosine similarity, as discussed in Section 5.3, are also relevant here. In the NMT literature, following bitext mining, which incorporates relative scores, an absolute scoring mechanism, such as a final cosine similarity threshold, is often applied to retain only high-quality pairs. Additionally, margin-based approaches have consistently shown better performance than standard cosine similarity methods (see: https://aclanthology.org/2021.acl-long.507/). Adding an absolute component to the formulation, would be beneficial.

2. The authors report correlation scores but omit absolute scores. Including absolute scores, at least for select tasks, would provide a more comprehensive evaluation and further strengthen the paper.

3. The authors’ evaluation is restricted to discriminative tasks, omitting open-ended text generation tasks. It is therefore unclear whether MEXA scores correlate with the LLM's multilingual text-generation capabilities.

4. FLORES (derived from Wikipedia) and BIBLE (potentially a component of the pre-training data) are used as parallel datasets; and many benchmarks may also be Wikipedia-based, however, it remains uncertain whether MEXA is generalizable in terms of demonstrating reliable correlations with other parallel datasets from unseen distributions. Further, the authors do not provide any method of determining the optimal choice of the parallel data to be utilized in the MEXA score computation.

5. The paper’s ablations are somewhat conventional, with previous studies having compared similar design choices. While the plots offer some valuable insights, the authors only discuss these superficially and omit in-depth analysis. For instance, since Belebele derives from FLORES, one would ideally expect it to exhibit a superior correlation with the prediction of downstream performance on Belebele. However, as illustrated in Figure 5.2, the Bible dataset exhibits a stronger correlation. The authors should explore and discuss such deviations from expected outcomes in their paper. Another example is the contrast between romanized versus native script performance, something that is actively being explored in literature, (see: https://aclanthology.org/2024.acl-long.833/, https://arxiv.org/abs/2201.12501). While the authors briefly discuss this aspect, they do not extend their analysis to other languages. Such detailed insights would add substantial value and enhance the overall assessment of the paper.

**Questions:**

1. How is the MEXA score for English in the first row of Table 1 calculated, given that the MEXA formulation is designed to compute alignment between English and a non-English language?

2. Prior works (https://arxiv.org/abs/2405.14782) have demonstrated that likelihood based evaluation and generative evaluation exhibit different trends, how would MEXA correlations change when switched to a generative evaluation?

3. Since the distribution of MEXA scores may vary depending on the model, the statement in lines 431-432 may not be appropriate to make.

4. How does one compare MEXA scores across different translation datasets to determine whether the trends remain consistent? If they are inconsistent, what criteria should guide the selection of the most appropriate translation dataset?

5. For different models, what is the optimal layer to compute MEXA scores? Does the choice of layer vary significantly between models, and for a new model, would it be necessary to conduct an independent analysis to determine the optimal layer, or could insights from existing experiments offer a reliable estimate?

6. Echo embeddings would serve as a valuable baseline and a useful ablation in this analysis (see: https://arxiv.org/abs/2402.15449).

---

> ### Author Response · Authors · 2024-11-25
>
> Thank you for your thoughtful review of our paper. The review is very thorough and completely understands the strengths of our paper (“problem is very well motivated”, “proposed method is straightforward”, “paper is well-structured”, “exhaustive experiments across a wide range of languages and models”); however, we did not expect this review score from this review.
>
> We hope we have addressed the reviewer’s questions so they can reconsider and provide a better score. We will address the issues raised in the revised version.
>
> **Weaknesses:**
>
> > 1) The MEXA formulation relies on relative scoring and does not account for the magnitude of cosine similarity ... Adding an absolute component to the formulation, would be beneficial.
>
> In all setups, with the same configurations and without exception, MEXA achieves better results than absolute cosine similarity. The idea suggested by the reviewer seems interesting, and we need to further investigate it. However, we are afraid that adding such a component would make the MEXA score not comparable between different models per language.
>
>
> > 2) The authors report correlation scores but omit absolute scores. Including absolute scores, at least for select tasks, would provide a more comprehensive evaluation and further strengthen the paper.
>
> For Llama 3.1 8B, for two tasks and two parallel datasets shown in Figure 1, we included the MEXA scores. In Appendix A.5, we also include MEXA scores for additional models on FLORES-200.
>
> > 3) The authors’ evaluation is restricted to discriminative tasks, omitting open-ended text generation tasks. It is therefore unclear whether MEXA scores correlate with the LLM's multilingual text-generation capabilities.
>
>
> The scope of this paper is limited to non-generative tasks. Generation is generally more challenging than understanding, and it is unsurprising that many languages may struggle to generate content in their language. While NLP has advanced toward generative capabilities, a significant portion of evaluation still focuses on non-generative tasks (e.g., [Hugging Face Leaderboards](https://huggingface.co/docs/leaderboards/en/open_llm_leaderboard/archive)) due to their convenience and standardized metrics.
> Assessing generated output remains challenging, even in English benchmarks. For example, model-based approaches (e.g., "LLM-as-a-judge") require an LLM fully competent in the target language—a capability that is both questionable and the focus of our evaluation.
>
>
> > 4) FLORES (derived from Wikipedia) and BIBLE (potentially a component of the pre-training data) are used as parallel datasets; and many benchmarks may also be Wikipedia-based, however, it remains uncertain whether MEXA is generalizable in terms of demonstrating reliable correlations with other parallel datasets from unseen distributions. Further, the authors do not provide any method of determining the optimal choice of the parallel data to be utilized in the MEXA score computation.
>
> For FLORES, only the English part is derived from Wikipedia, while the rest of the languages are human-translated. In the case of the Bible, some models tend to memorize the English portion. We demonstrate that in both scenarios, using the weighted average embedding approach, we can achieve generalizable scores without concerns about memorization. We believe that testing on more unseen distributions (we would gladly consider any unseen parallel dataset recommendations from the reviewer) could yield lower MEXA scores. However, this only illustrates how the model performs on unseen distributions in other languages and does not undermine the validity of the MEXA approach.
>
>
> > 5) The paper’s ablations are somewhat conventional, with previous studies having compared similar design choices. While the plots offer some valuable insights, the authors only discuss these superficially and omit in-depth analysis.
>
> We will reference the suggested papers when discussing romanization. Additionally, we will expand the discussion of the findings in the figures.

---

> > ### Author Response · Authors · 2024-11-25
> >
> > **Questions:**
> >
> > > 1) How is the MEXA score for English in the first row of Table 1 calculated.
> >
> > In Table 1, we do not provide any MEXA scores for English. The first row represents the task itself, showing how well, for example, Beleble performs in English. The MEXA scores are the last four rows. We will add more details to the caption for clarity.
> >
> >
> > > 2) Prior works (https://arxiv.org/abs/2405.14782) have demonstrated that likelihood based evaluation and generative evaluation exhibit different trends, how would MEXA correlations change when switched to a generative evaluation?
> >
> > The scope of this paper is limited to non-generative tasks; understanding naturally comes before generation, and different trends are to be expected. However, since we have not yet identified a generative benchmark that is parallel and supports enough languages for comparison with MEXA, we do not make any claims.
> >
> > > 3) Since the distribution of MEXA scores may vary depending on the model, the statement in lines 431-432 may not be appropriate to make.
> >
> > MEXA itself is a downstream task (bidirectional retrieval P@1), in contrast to absolute cosine similarity. It's comparable between models, and the variation across models shows the extent of performance variation.
> >
> > > 4) How does one compare MEXA scores across different translation datasets to determine whether the trends remain consistent? If they are inconsistent, what criteria should guide the selection of the most appropriate translation dataset?
> >
> > By translation datasets, we assume the reviewer means parallel datasets. We tested two extensive ones that cover many languages in this paper. The important thing is that the datasets should be as unmemorized as possible so the embeddings are more reliable, even though the weighted embeddings can address this issue.
> >
> >
> > > 5) For different models, what is the optimal layer to compute MEXA scores?
> >
> > There is no single optimal layer; it can vary depending on the model and language. For some languages and models, the last layer or the middle layers are sometimes the best. We did not select additional hyperparameters that could bias our results. Instead, we used two general pooling methods to provide generalized solutions for new models and tasks.
> >
> >
> > > 6) Echo embeddings would serve as a valuable baseline and a useful ablation in this analysis (see: https://arxiv.org/abs/2402.15449).
> >
> >
> > We took the reviewer’s suggestion and tested echo embeddings to retrieve embeddings on Llama 3.1 8B using the last token. To our surprise, we found that using just one sentence in most scenarios is noticeably or equally better than echo embeddings. There is also not much difference in the original paper on other embedding tasks (Echo vs. Classical last token).

---

> > > ### Author Response · Authors · 2024-11-27
> > >
> > > Dear Reviewer,
> > >
> > > Thank you once again for reviewing our paper. We kindly ask for your consent to confirm whether the author responses have adequately resolved the issues you raised. You initially provided a very detailed and positive review of the paper, assigning scores of 3 for soundness, presentation, and contribution. However, the overall rating was 5. We hope our answers address your concerns to reconsider the score.
> > >
> > > We would be happy to address any additional questions or suggestions you may have.

---

### Official Review · Reviewer_QBjb · 2024-11-05

**Soundness:** 2
**Presentation:** 2
**Contribution:** 2
**Rating:** 5
**Confidence:** 3

**Summary:**

The paper presents a method, MEXA, to evaluate the multilingual strength or capability of English centric LLMs. MEXA is an intrinsic evaluation method that utilizes a parallel corpus and calculate the embedding distances between parallel sentences in comparison to two random sentences in two different languages. A number of evaluations are conducted using several 7B models and using approximately 100 parallel examples from two datasets. The results show that their proposed method strongly correlates with the multilingual performance of the under-observation models.

**Strengths:**

The approach is simple and empirical results are quite strong.

**Weaknesses:**

I put my questions and weaknesses in the questions section and would encourage authors to respond to them.

**Questions:**

- The authors use only 100 examples in their evaluation which is a very small number while they have more examples available. Even if they want to keep the compute small, they should try a few models on more examples and may show that their approach require a little as 100 parallel examples.

- The authors mention several tasks in Appendix but then decided to use only three tasks. I did not understand the relation behind this choice and why authors did not try to run of a larger number of tasks to show the generalization of their methodology. Authors also mentioned that the primary focus of their paper is natural understanding task. As far as I understood, their technique is general and should work for a diverse set of tsaks.

- Presentation of results in tables make it very hard to digest them. Authors may select one best strategy like mean/max and present the results. Table 1 and 2 both are very hard to understand.

- What is the motivation of keeping the proposed method limited to monolingual models? What about multilingual models? Expanding the experiments to multilingual model will greatly increase the impact and breadth of the proposed method.

- I did not understand which of the final combination of layers work the best for MEXA? Is it a combination of all layers or a specific layer? From Figure 2, it seems that the alignment scores are best for middle layers. Is it correct? What is the optimal layer for MEXA?


- A few arguments need empirical support or support from literature. - Line 50-51: for these models to be effective in other languages, it is important that the other languages align with the main language. The basis of this assumption is unclear and authors should support it empirically or from literature.

- What is the exact prompt given to models and what is their impact?

- Figures and tables are extremely small. I was unable to read them on paper. I am not sure this small size is even allowed by the conference. Authors should move a few models to appendix and present fewer models.

---

> ### Author Response · Authors · 2024-11-25
>
> Thank you for your valuable feedback on our paper. We will incorporate your suggestions in the revised version.
>
>
> **Questions:**
>
> > 1) The authors use only 100 examples in their evaluation which is a very small number while they have more examples available.
>
> Thanks for pointing this out! Initially, we used 1000 sentences from the FLORES dataset for Llama series evaluations and found only a small difference (±1%) in correlations when reduced to 100. In Appendix 4, we show that 100 samples are sufficient to ensure MEXA's robustness, with high pearson scores validating this. Using 100 samples also allows scaling to more languages, many of which lack 100+ parallel samples. We will include a discussion of the small difference observed with larger batches in the paper.
>
>
> > 2) The authors mention several tasks in Appendix but then decided to use only three tasks.
>
>
> We include all the possible tasks in Table 4 to explain why we chose these tasks. As you can see, for non-generative tasks, only these tasks support a high number of languages, including low-resource ones. Benchmarks with only 10 languages, mostly high-resource, would not support our claims, as MEXA would achieve high results for all of them. Belebele is the benchmark with the highest number of languages, while ARC and MMLU are the most important tasks on the community leaderboards for comparing models.
>
>
> > 3) Presentation of results in tables make it very hard to digest them. Authors may select one best strategy like mean/max and present the results. Table 1 and 2 both are very hard to understand.
>
>
> Thanks for the suggestion! As explained in L381, mean and max pooling don’t have an advantage over each other but capturing different levels of understanding, both of which are important. To simplify Table 2, we will present only the average results and will reformat the table, moving the remaining numbers to the appendix for clarity.
>
>
> > 4) What is the motivation of keeping the proposed method limited to monolingual models?
>
>
> The models discussed in this paper are certainly not monolingual. For example, if the pretraining data contains 0.1% of one non-English language and the total training data consists of 3 trillion tokens, this equates to 3 billion tokens in that language. However, the reason these models are English-centric is that MEXA is based on the premise that the model has one latent language, and performance in other languages can be predicted by their level of alignment with the model's latent language.
>
>
> > 5) I did not understand which of the final combination of layers work the best for MEXA? Is it a combination of all layers or a specific layer? From Figure 2, it seems that the alignment scores are best for middle layers. Is it correct? What is the optimal layer for MEXA?
>
> There is no single optimal layer; it can vary depending on the model and language. For some languages and models, the last layer or the middle layers are sometimes the best. We did not select additional hyperparameters that could bias our results. Instead, we used two general pooling methods to provide generalized solutions for new models and tasks.
>
>
> > 6) A few arguments need empirical support or support from literature. - Line 50-51: for these models to be effective in other languages, it is important that the other languages align with the main language. The basis of this assumption is unclear and authors should support it empirically or from literature.
>
>
> In an English-centric LLM, the main language drives better performance, reasoning, and knowledge, while most other languages suffer from insufficient data relative to the main language. It is logical to say, in this scenario, if there is no alignment between the main language and other languages, the model will not provide meaningful coverage for the other languages.
>
>
> > 7) What is the exact prompt given to models and what is their impact?
>
> For MEXA, no prompt templates are given to the models, only the parallel sentences exactly as they are. For the tasks, we use the template exactly as in the lm-evaluation-harness.
>
>
> > 8) Figures and tables are extremely small. I was unable to read them on paper. I am not sure this small size is even allowed by the conference. Authors should move a few models to appendix and present fewer models.
>
> We will be sure to do this for the model tables and ensure that the figures are larger. We will move the similar models in performance and represent only one of each kind in the main text.
>
>
> **\*\*** We appreciate the reviewer's feedback and most of these suggestions are simple fixes that we would certainly apply to final revision. Based on the strengths, (refer to the general response), we hope the reviewer can reconsider and provide a better score.

---

> > ### Author Response · Authors · 2024-11-28
> >
> > We kindly ask the reviewer if the authors' response has resolved the questions you raised. We would be happy to address any additional questions or suggestions you may have. Thank you!

---

> > > ### Comment · Reviewer_QBjb · 2024-12-02
> > > **Thank you**
> > >
> > > My apology for the late response.
> > >
> > > I read the author's response to all reviewers comments and also went through all of the reviews.
> > >
> > > - In response to all reviewers' comments, authors should have made an attempt to update the paper by using the flexibility provided by ICLR. The responses saying that they will fix suggestions in the final version is not sufficient in my opinion.
> > >
> > > - I am confused on the applicability of MEXA based on response 2 and 4. Authors mentioned that for high-resource languages, MEXA would achieve high results for all languages. Then point 4 mentioned that MEXA is based on the premise that the model has one latent language. What is the basis of this assumption? It is unclear why MEXA can not use to intrinsically evaluate the performance of multilingual models for various language pairs.
> > >
> > > - I am unclear on author's response on not using generative tasks. Since MEXA works at embedding level and does not rely on prediction, why would it make it hard to apply on generative tasks? What about giving an English sentence as input and then a foreign language sentence as input?

---

> > > > ### Author Response · Authors · 2024-12-02
> > > >
> > > > Thanks for the feedback.
> > > >
> > > > > In response to all reviewers' comments, authors should have made an attempt to update the paper ...
> > > >
> > > > We appreciate the reviewer's feedback. However, most of these suggestions are simple fixes (1. We will include this exact discussion; 2 to 7 are questions answered for clarity; and 8 involves revising the representation of the table, which we have included in our plan) that we already addressed in the rebuttal.
> > > >
> > > > > I am confused on the applicability of MEXA based on response 2 and 4 ...
> > > >
> > > > Our paper presents a hypothesis that such models possess an underlying latent language structure, e.g, English (based on https://arxiv.org/abs/2402.10588). We test the hypothesis that the alignment of embeddings for other languages with English across different layers can be used to assess the multilingual capabilities of these models. Based on the "empirical results are quite strong" findings, we conclude that our hypothesis is supported.
> > > >
> > > > > I am unclear on author's response on not using generative tasks ...
> > > >
> > > > While NLP has advanced toward generative capabilities, a major portion of evaluation still focuses on multiple-choice tasks (e.g., [Hugging Face Leaderboards](https://huggingface.co/docs/leaderboards/en/open_llm_leaderboard/archive)) due to their convenience and standardized metrics. Assessing generated output remains challenging, even in English benchmarks! For example, model-based approaches (e.g., "LLM-as-a-judge") require an LLM fully competent in the target language—a capability that is both questionable and the focus of our evaluation.
> > > >
> > > > The other reason is that we could not find a suitable parallel benchmark to compare MEXA on generative tasks (Table 4). We aimed to design a method based on the idea of generating text in all languages using each model (which would provide LLM-generated text in parallel) and using these new parallel outputs to apply MEXA. However, the inability to compare this idea limits the scope of this paper to non-generative tasks. Generation is generally more challenging than understanding, so it is unsurprising that many languages may face difficulties generating content in their native language.
> > > >
> > > >
> > > > ** We hope we have addressed your concerns and would be happy to respond to any additional questions or suggestions you may have.

---

### Official Review · Reviewer_HipV · 2024-11-06

**Soundness:** 3
**Presentation:** 4
**Contribution:** 2
**Rating:** 6
**Confidence:** 4

**Summary:**

In this paper, the authors propose a method to predict the performance of English-centric LLMs on other languages using information about the alignment between English and other languages in the LLMs. The central hypothesis here is that the LLMs performance on a task is primarily dependent on the English performance, and performance on other languages depends on how well its representations are aligned with English. The paper proposes a measure MEXA to compute the similarity of representations between languages and shows that MEXA score is high correlated to the language performance on a task. Results are shown on 3 classification tasks and multiple, diverse languages including low-resource languages. MEXA score only requires a small parallel corpus to compute the alignment scores between the languages.

**Strengths:**

- The paper proposes a simple approach for prediction of multilingual performance, that requires few resources and is easy to implement.
- The experiments are done on a wide range of languages, showing inclusivity and making the results solid.
- The related work on understanding alignment is well-covered.

**Weaknesses:**

- There is prior work on prediction of performance of multilingual NLP models based on various features like linguistic similarity, vocab overlap, embedding similarity, pre-training data size etc. These have not been discussed. This is directly relevant to the work, and in fact the proposed work can be considered a special case of many of these approaches, where only a single feature - the MEXA score is used for performance prediction. In the light of these works, the novelty is limited. The moot question to answer is how good is just a single feature - the MEXA score compared to using a plethora of features that previous works have used. A discussion with respect to previous work would be useful.
- The work focusses only on classification tasks, and does not cover generation tasks. When the NLP landscape have moved towards generative capabilties, evaluating prediction of generation quality is also important. This has been mentioned as a limitation of the work in the paper.
- The comparison with absolute cosine similarity is very limited (Section 5.3). To solidly establish the benefits of MEXA over cosine similarity, it would be good to report absolute cosine similarity results with the same default settings (weighted average, mean pooling) across all tasks and models.

References:

1. Ahuja, Kabir, Shanu Kumar, Sandipan Dandapat, and Monojit Choudhury. "Multi task learning for zero shot performance prediction of multilingual models." ACL. 2022.
2. Ye, Zihuiwen, Pengfei Liu, Jinlan Fu, and Graham Neubig. "Towards more fine-grained and reliable NLP performance prediction." EACL. (2021).
3. Xia, Mengzhou, et al. “Predicting Performance for Natural Language Processing Tasks.” Proceedings of the 58th Annual Meeting of the Association for Computational Linguistics, Association for Computational Linguistics, 2020, pp. 8625–46. Crossref, https://doi.org/10.18653/v1/2020.acl-main.764.
4. Srinivasan, Anirudh, Sunayana Sitaram, Tanuja Ganu, Sandipan Dandapat, Kalika Bali, and Monojit Choudhury. "Predicting the performance of multilingual nlp models." arXiv preprint arXiv:2110.08875 (2021).

**Questions:**

Table 2 is complicated to read. Please consider reorganizing to improve readability.

---

> ### Author Response · Authors · 2024-11-25
>
> Thank you for your positive review of our paper! We appreciate your suggestions and will address them in the updated version. Thanks!
>
> **Weaknesses:**
>
> > 1) There is prior work on prediction of performance of multilingual NLP models ...
>
> True, prior work has explored features such as linguistic similarity, vocabulary overlap, embedding similarity, and pretraining data size, primarily in encoder-only models. However, these features often require access to pretraining data, which is unavailable for open-weight LLMs. This paper focuses on MEXA, a feature derived from the pre-trained model, without relying on pretraining data details. The novelty lies in demonstrating that a single representational-based measure, MEXA, is sufficient for performance estimation in English-centric LLMs. Thank you for pointing this out. We will include the suggested papers and incorporate this discussion to better differentiate our work in the paper.
>
>
> > 2) The work focusses only on classification tasks, and does not cover generation tasks. When the NLP landscape have moved towards generative capabilties, evaluating prediction of generation quality is also important. This has been mentioned as a limitation of the work in the paper.
>
>
> Yes, the scope of this paper is limited to non-generative tasks. Generation is generally more challenging than understanding, and it is unsurprising that many languages may struggle to generate content in their language. While NLP has advanced toward generative capabilities, a major portion of evaluation still focuses on multiple-choice tasks (e.g., [Hugging Face Leaderboards](https://huggingface.co/docs/leaderboards/en/open_llm_leaderboard/archive)) due to their convenience and standardized metrics.
> Assessing generated output remains challenging, even in English benchmarks. For example, model-based approaches (e.g., "LLM-as-a-judge") require an LLM fully competent in the target language—a capability that is both questionable and the focus of our evaluation.
>
>
>
> > 3) The comparison with absolute cosine similarity is very limited (Section 5.3) ...
>
> In all of the setups, with the same configurations, without exception, MEXA achieves better results than absolute cosine similarity. We have included the comparison with the settings that were already explored by previous works in the main text. The remaining comparisons will be included in the appendix.
>
> **Questions:**
>
> > 1) Table 2 is complicated to read ...
>
> Sure, we will move the detailed results to the appendix and only keep the different settings with average results in the main paper.

---

### Author Response · Authors · 2024-11-25
**General Response**

We thank the reviewers for their positive reviews and helpful comments regarding the:

- **problem is well-motivated** (i2iS: “The problem is very well motivated, particularly given the lack of multilingual benchmarks”)
- **method novelty, simplicity, and strong empirical results** (QBjb:“The approach is simple and empirical results are quite strong.”, oR5Z:“It is impressively accurate at predicting the performance of several decoder-only English-centric LLMs”, bpzp: “The main finding ... is robustly supported by the results”, i2iS: “The proposed method is straightforward”)
- **thorough evaluations** (bpzp: “The authors have explored a number of design decisions for Mexa ... and present a full analysis of their finding“,  HipV: “experiments are done on a wide range of languages, showing inclusivity and making the results solid”, i2iS: “The authors perform exhaustive experiments across a wide range of languages and models, offering valuable, in-depth insights.”)
- **well-structured paper** (i2iS: “The paper is well-structured and easy to follow.”)
- **well-covered related work** (HipV: “The related work on understanding alignment is well-covered.”)
- **contribution to the field and inspiring future works** (bpzp: “The study ... produce new insights into the mechanisms underlying cross-lingual transfers in LLMs.”, oR5Z: “I could see this approach inspiring future work in analysis of multilingual LLM performance.”)
- **ease of reproducibility** (HipV: “is easy to implement”)

We will make sure to include all the suggestions provided by the reviewers as promised for the revised version.

---

> ### Author Response · Authors · 2024-12-01
>
> Dear Reviewers,
>
> Thank you again for reviewing the MEXA paper; we truly appreciate your efforts. We have taken the time to address all of your questions. As of now, **none** of the reviewers have responded to indicate whether our comments resolve the issues you raised. We understand you may be busy on your time, but we would be grateful if you could take a moment to provide your feedback on the author responses—it would make a significant difference. Thank you for considering!
>
> December 2 is the final day reviewers may post a message to the authors. Given this [recommendation](https://x.com/abeirami/status/1861896504732008481), it would be useful to articulate answers to the following questions in 1-2 sentences each when responding to the author rebuttal:
>
> 0. Does the response from the authors resolve the issues you raised?
> 1. Why not a lower score?
> 2. Why not a higher score?

---

### Author Response · Authors · 2024-12-04
**Rebuttal Summary by Authors**

This paper received five reviews: two are positive, one is borderline (good sub-scores, bad total score), and two are more negative.

Positives:
- bpzp (score: 6, confidence: 5, soundness: 4, presentation: 3, contribution: 3)
- HipV (score: 6, confidence: 4, soundness: 3, presentation: 4, contribution: 2)

Borderline:
- i2iS (score: 5, confidence: 4, soundness: 3, presentation: 3, contribution: 3)

More negative:
- oR5Z (score: 5, confidence: 4, soundness: 2, presentation: 3, contribution: 3)
- QBjb (score: 5, confidence: 3, soundness: 2, presentation: 2, contribution: 2)


We believe the reviewer bpzp (confidence: 5) provided a very accurate review of our paper. Among the reviews, we had hoped that reviewer i2iS (confidence: 4) would respond to confirm if our answers addressed the questions they raised, as their review is very positive and includes good scores (soundness: 3, presentation: 3, contribution: 3). However, they did not respond so their decision score remains 5. Among all the reviewers, only oR5Z and QBjb responded to us, but their decisions did not change.

The oR5Z final comment states: “my concerns about the usefulness of Mexa and the use of machine-translated data have not changed”
- For the usefulness of MEXA, we refer to the general response, in which we gathered the strengths of the paper.
- We would clarify that our method, MEXA, does not use any machine-translated data. In fact, we rely on human-translated data (FLORES and Bible) for our experiments. However, to demonstrate that MEXA scores align with the current standards of multilingual evaluation, we needed to compare them with popular machine-translated benchmarks, which are representative of the current state of automatic multilingual evaluation.

Reviewer QBjb has a much lower confidence score (confidence: 3) compared to other reviews, and we believe they did not fully grasp the objective of the paper. Their review primarily focused on simple fixes (they raised eight questions; we responded to seven, and for the last question, they requested a better representation of a table which we promised). However, it seems they remain unconvinced.

---

### Note · Authors · 2025-02-12

I have read and agree with the venue's withdrawal policy on behalf of myself and my co-authors.

---

### Meta-Review · Area_Chair_m8wt · 2024-12-19

**Metareview:**

The paper proposes Mexa, a method to assess the multilingual capabilities of English-centric large language models (LLMs) by evaluating the alignment of representations between English and other languages using parallel datasets. Mexa aims to predict task performance across multiple languages without relying on pretraining data or multilingual benchmarks, which are scarce for low-resource languages. The results demonstrate a strong correlation between Mexa scores and downstream task performance, achieving a Pearson correlation of 0.90 on average across nine models and two parallel datasets. Experiments are performed on decoder-only models, specifically the Llama, Gemma, Mistral, and OLMo families. The paper concludes that Mexa provides insights into cross-lingual transfer mechanisms and serves as a practical, resource-efficient measure for estimating multilingual performance.

The paper’s strengths lie in its motivation, simplicity, and comprehensiveness of experiments. The authors address an important problem of evaluating multilingual performance for English-centric LLMs, especially for low-resource languages. The approach is lightweight, requiring only parallel data, and achieves significant correlation with multilingual task performance. The experiments are exhaustive across languages and models, offering practical and valuable insights. The paper is also well-structured, making the contributions accessible to the broader NLP community.

However, several critical weaknesses limit the paper's overall contribution. First, the scope of evaluation is narrow, as the authors focus exclusively on classification tasks, ignoring generative tasks. Despite claiming the method’s generalizability, its applicability to text generation remains unclear. Second, the paper lacks sufficient comparison with absolute cosine similarity and other existing metrics for performance prediction. Although Mexa performs better, the evidence provided is incomplete. Third, the choice of parallel datasets (FLORES and Bible) raises concerns about generalizability. The FLORES dataset heavily relies on Wikipedia-based translations, while the Bible dataset may be partially memorized by LLMs. The paper fails to explore datasets from unseen distributions, which undermines the robustness of their claims. Fourth, the focus on decoder-only, open-weight LLMs limits Mexa's practical applicability, as closed-source models cannot be evaluated. Furthermore, the assumption that English serves as a latent pivot language across layers is presented without sufficient empirical support or justification, potentially reducing the generalizability of the findings. Finally, the presentation of results is overly complex and difficult to interpret, as highlighted by reviewers.

Given these limitations, the contributions of Mexa, though interesting, are incremental relative to prior work on predicting multilingual performance using alignment metrics or embedding similarity. While the authors provided detailed rebuttals, most responses were promises to address weaknesses in a revised version rather than concrete updates, and critical concerns remain unaddressed. Due to the limited novelty of the method, combined with the restricted evaluation scope and concerns about generalizability, the paper does not meet the bar for acceptance at ICLR. Thus, I recommend rejecting the paper in its current form.

**Additional Comments On Reviewer Discussion:**

During the rebuttal period, the discussion centered on three primary concerns: scope of evaluation, generalizability, and novelty. Reviewers raised significant questions regarding these points, and while the authors provided clarifications, they did not fully resolve the issues.
1. Scope of Evaluation: Reviewers, particularly HipV and i2iS, highlighted the paper's exclusive focus on classification tasks and its omission of generative tasks, which are increasingly central in NLP. The authors argued that generative benchmarks are less standardized and harder to evaluate across languages, but this justification was unconvincing.

2. Generalizability: Multiple reviewers (QBjb and oR5Z) raised concerns about the choice of datasets, suggesting that these might not generalize well to unseen distributions. Reviewers pointed out FLORES's reliance on Wikipedia and the risk of memorization for the Bible dataset. The authors responded that weighted embeddings mitigate memorization issues and welcomed recommendations for additional datasets, but they did not test other distributions. This leaves the question of generalizability unresolved.

3. Novelty and Comparisons: HipV and bpzp emphasized the incremental nature of Mexa, noting that it builds on prior alignment-based performance prediction methods without introducing substantial new insights. Mexa’s reliance on straightforward representational comparisons was seen as lacking novelty. Furthermore, reviewers questioned the limited comparison with baseline metrics like absolute cosine similarity. The authors asserted that Mexa consistently outperforms cosine similarity but provided no comprehensive evidence across all settings, leaving gaps in their argument.

4. Applicability and Model Coverage: Reviewer oR5Z criticized the focus on open-weight, decoder-only models, arguing that Mexa’s utility is constrained without evaluation on closed-source or other architectures. The authors defended this choice as aligned with their focus on open, English-centric models, but the restriction significantly limits Mexa’s practical use in real-world scenarios.

In my final decision, I weighed all these points. While the authors’ responses clarified some technical details, they primarily deferred addressing key weaknesses (e.g., dataset generalizability, scope of evaluation) to a future revision. The omission of generative tasks and the narrow dataset and model focus remain major limitations that undermine the paper's broader impact. Additionally, the lack of substantial novelty and limited comparisons to existing baselines weaken the significance of the contribution. Given these unresolved concerns, I concluded that the submission does not meet the standards for acceptance at ICLR.

---

### Decision · Program_Chairs · 2025-01-22

Reject